# Mitigating Preference Hacking in Policy Optimization with Pessimism

## Abstract

This work tackles the problem of overoptimization in reinforcement learning from human feedback (RLHF), a prevalent technique for aligning models with human preferences. RLHF relies on reward or preference models trained on *fixed preference datasets*, and these models are unreliable when evaluated outside the support of this preference data, leading to the common reward or preference hacking phenomenon. We propose novel, pessimistic objectives for RLHF which are provably robust to overoptimization through the use of pessimism in the face of uncertainty, and design practical algorithms, P3O and PRPO, to optimize these objectives. Our approach is derived for the general preference optimization setting, but can be used with reward models as well. We evaluate P3O and PRPO on the tasks of fine-tuning language models for document summarization and creating helpful assistants, demonstrating a remarkable resilience to overoptimization.

## 1 Introduction

Reinforcement learning (RL) from human feedback (RLHF) (Christiano et al., 2017) has emerged as a promising technique for aligning language models with human preferences (Stiennon et al., 2020; Ouyang et al., 2022). The predominant approach involves training a reward model on human preference data and then fine-tuning the language model to maximize the expected reward of the responses it generates to training inputs. More recently, a line of work (Swamy et al., 2024; Munos et al., 2023; Calandriello et al., 2024; Guo et al., 2024) has argued for the benefits of learning a *pairwise* preference function from the preference dataset, and using this to directly compare trajectories side-by-side during online RL. Irrespective of whether we use reward or preference models during training, however, the availability of a limited pool of high-quality preference data presents a key bottleneck in learning good policies. The high cost of collecting preference datasets with human feedback means that they suffer from limited coverage, and the reward or preference models trained on such datasets fail to adequately generalize to policies which produce trajectories out of the support of the preference data.

The inadequacy of learned reward/preference models in reliably producing good policies has resulted in the now well-documented phenomenon of reward hacking or overoptimization (Amodei et al., 2016; Gao et al., 2023; Eisenstein et al., 2024). The most commonly used technique to limit overoptimization is through regularization by the KL-divergence between the policy being trained and a reference model. However, KL-divergence only measures the distributional distance of the produced responses, independently of the uncertainty in the predictions of the reward/preference models. This typically leads to either overoptimization or overly limiting reward/preference maximization, when too little or too much regularization is applied, necessitating careful tuning. Even with careful tuning, KL regularization still often inhibits learning.

Motivated by these concerns, there is a growing literature on techniques to control this overoptimization behavior in more data-driven ways, such as by incorporating uncertainty in the predictions of the underlying reward model with explicit reward ensembles (Eisenstein et al., 2024; Coste et al., 2023), or pessimistic reasoning (Fisch et al., 2024; Liu et al., 2024; Huang et al., 2024b; Cen et al., 2024), albeit with only modest success in standard settings. Similar to the related area of distributionally robust optimization Bertsimas & Sim (2004); Ben-Tal et al. (2013), a core part of the challenge is balancing sufficient reward uncertainty and

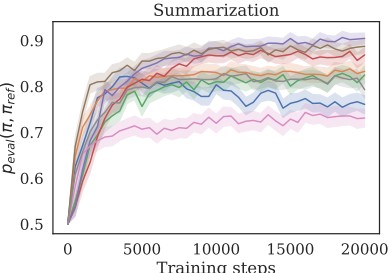
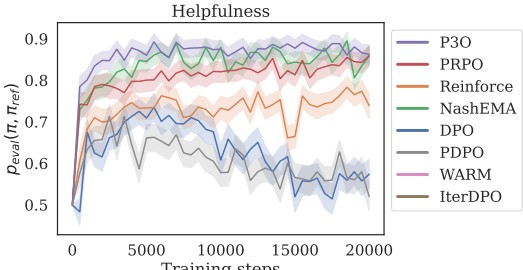

Figure 1: Comparison of our methods (P3O and PRPO) with standard approaches (REINFORCE Williams (1992), NashEMA Munos et al. (2023), DPO Rafailov et al. (2023)), and additional variations thereof (IterDPO, WARM Ramé et al. (2024)), including pessimistic extensions by prior work (PDPO Fisch et al. (2024)). Evaluations are shown for summarization and "helpful assistant" (helpfulness) tasks. The vertical axis shows preference win-rates from a prompted Gemini evaluator, comparing each method's generations against a reference policy. All methods are hyperparameter-tuned for best eval performance while avoiding reward hacking, with methods like DPO, NashEMA, and REINFORCE all requiring strong KL regularization. In contrast, our methods, P3O and PRPO, avoid reward hacking without relying on KL regularization—instead using a novel form of pessimism—and consistently obtain higher eval scores. While IterDPO and NashEMA perform competitively on the summarization and helpfulness tasks, respectively, both tend to produce longer, more idiosyncratic outputs (see Figure 3 and Table 1), whereas P3O and PRPO are able to maintain more natural generation patterns. Shaded regions denote 95% confidence intervals.

pessimism to prevent overoptimization, while still learning effectively Eisenstein et al. (2024); Fisch et al. (2024).

In contrast to the reward-based setting, much less work has studied the incorporation of uncertainty when using learned pairwise preference models in subsequent RL—with even fewer works on good definitions of uncertainty in this setting. Pessimistic techniques for preference-based RL are particularly challenging, since many leading non-pessimistic methods (Munos et al., 2023; Swamy et al., 2024; Calandriello et al., 2024) critically leverage the symmetry of the min and max players to develop efficient algorithms, but the introduction of pessimism breaks this symmetry. Additionally, prior theoretical works Cui & Du (2022); Ye et al. (2024) exhibit some pathologies when the offline dataset has gaps in its coverage, due to the non-transitivity of general preferences. Consequently, their pessimistically optimal policies do not compete with policies whose responses are adequately covered in the data—the typical benchmark for offline RL in reward-based settings.

In this work, we build on prior works in preference-based RLHF (Swamy et al., 2024; Munos et al., 2023), and offline learning in Markov games (Cui & Du, 2022), to obtain new robust objectives for incorporating a *restricted* form of uncertainty from finite preference datasets. Specifically, we make the following key contributions:

1. We identify drawbacks of prior pessimistic estimators (Cui & Du, 2022; Ye et al., 2024) in the absence of prohibitive coverage assumptions on the preference data sampling policy. We then develop a new *restricted Nash formulation* under which the learned policy is provably preferable to any other policy that is restricted to choosing actions within the support of the preference data sampling policy.

2. We provide a practical algorithm, Pessimistic Preference-based Policy Optimization (P3O), for optimizing the resulting objective. We approximate the ideal theoretical objective with a variational upper bound, that yields a minimax game between a policy and a preference player, which we solve using gradient ascent-descent. The policy optimization is similar to prior works (Swamy et al., 2024; Munos et al., 2023) and the preference updates are adversarial to the current policy's choices. For Bradley-Terry-Luce models with reward-based preferences, we also evaluate a simpler, reward-based variant of P3O (which we call PRPO) in our experiments.

3. Empirical evaluation on document summarization and training helpful assistants in Figure 1 shows P3O and PRPO reach a higher quality of responses quickly, and the quality does not degrade due to overoptimization from further training. This is contrast with standard RLHF methods (DPO using no explicit reward/preference models (Rafailov et al., 2023), NashEMA using pairwise preference

models (Munos et al., 2023), and REINFORCE using pointwise reward models (Williams, 1992; Ahmadian et al., 2024)), which either exhibit significant overoptimization, or are limited in their ability to sufficiently optimize. The evaluations are performed using a prompted Gemini 1.5 Flash auto-evaluator (Gemini Team, 2024). Detailed analysis in Section 5 further shows that P3O and PRPO avoid the qualitative reward hacking behaviors of REINFORCE, NashEMA and DPO.

## 2  Background

We consider human alignment of a language model (LM) policy $\pi \in \Pi$, where $\Pi \subseteq \{\mathcal{X} \to \Delta(\mathcal{Y})\}$[1], which generates for a context $x \in \mathcal{X}$, a response $y \sim \pi(\cdot|x)$ with $y \in \mathcal{Y}$. We assume that we are given access to a preference dataset, $\mathcal{D}$, consisting of tuples $(x, y_w, y_l) \in \mathcal{X} \times \mathcal{Y} \times \mathcal{Y}$ where for context $x$, the response $y_w$ is preferred over $y_l$ (as labeled by a human). We further assume access to a reference policy $\pi_{\text{ref}}$, which may or may not match the original sampling policy for the preferred and dispreferred $(y_w, y_l)$ responses in $\mathcal{D}$. For brevity, we drop the context $x$ from the notation and work with a finite $\mathcal{Y}$ when there is no ambiguity.

Preferences are often modeled via a reward function under the Bradley-Terry-Luce (BTL) (Bradley & Terry, 1952; Luce, 2012) model (Christiano et al., 2017; Ouyang et al., 2022); however, in this paper, we make no such assumptions and work with both general pairwise preference functions as well as BTL preference functions based on pointwise rewards. In the following, we first set up the preference learning framework, and then discuss techniques to optimize policies with preference feedback, while also establishing the use of pessimism to handle uncertainties that may exist in the reward and preference functions.

**Learning preferences from data.**  We define the preference function $p : \mathcal{Y} \times \mathcal{Y} \to [0, 1]$, such that $p(y_1, y_2) \doteq \Pr(y_1 \succ y_2)$ represents the probability of the generation $y_1$ being preferred over $y_2$ by a target user. Being a probability, the preference function satisfies: $p(y_1, y_2) = 1 - p(y_2, y_1)$.[2] To obtain a preference model, we typically fine-tune a pretrained language model (LM) on $\mathcal{D}$ to produce the maximum likelihood estimate $p_{\text{mle}}$ via the following objective:

$$p_{\text{mle}} \in \arg\min_p \mathcal{L}_{\text{pref}}(p; \mathcal{D}), \quad \text{where} \quad \mathcal{L}_{\text{pref}}(p; \mathcal{D}) = -\mathbb{E}_{(y_w, y_l) \sim \mathcal{D}} \left[ \log p\left(y_w, y_l\right) \right]. \tag{1}$$

We overload the notation to say $p(\pi, \pi')$, where $\pi, \pi' \in \Pi$, to represent the expected preference for $\pi$ over $\pi'$, given the preference function $p$, that is, $p(\pi, \pi') = \mathbb{E}_{y \sim \pi, y' \sim \pi'}[p(y, y')]$.

In the standard case of RLHF, the preference function is modeled using a reward function $r : \mathcal{Y} \to \mathbb{R}$, assuming an underlying BTL model $p_{\text{BTL}}$, i.e.,

$$p_{\text{BTL}}(y_1, y_2; r) \doteq 1/(1 + \exp(r(y_2) - r(y_1))). \tag{2}$$

Note that the set of BTL models is a strict subset of general pairwise preference models. We overload the notation $p_{\text{BTL}}(r)$ to denote the BTL preference model that is induced by reward function $r$. Throughout the paper, we use $p^\star$ to denote the *ground-truth* preference function that reflects the true human preferences and from which the preference dataset $\mathcal{D}$ was generated. If $\mathcal{D}$ contains samples of the form $(x, y_w, y_l)$, then the implicit $p^\star$ can be written as $p^\star(y \succeq y'|x) = \mathcal{P}_D(y_w = y, y' = y_l|x)$, where $\mathcal{P}_\mathcal{D}$ denotes probabilities under the distribution $\mathcal{D}$.

**Preference-based policy optimization.**  To optimize general preferences without making a BTL modeling assumption, following Munos et al. (2023) and Swamy et al. (2024), we formulate a preference game $J_{\text{P}}(\pi, \pi', p)$ between a pair of competing policies $\pi$ and $\pi'$, with preference function $p$, a reference policy $\pi_{\text{ref}}$, and a regularization parameter $\beta > 0$, as

$$J_{\text{P}}(\pi, \pi', p) \doteq p(\pi, \pi') - \beta \text{KL}\left(\pi \| \pi_{\text{ref}}\right) + \beta \text{KL}\left(\pi' \| \pi_{\text{ref}}\right),$$

---

[1]We use $\Delta(\mathcal{Y})$ to denote the probability simplex defined over the elements of the set $\mathcal{Y}$ (e.g., the set of possible LM responses).
[2]While standard preference datasets typically assume strict preferences ($y_w \succ y_l$), ties are naturally accommodated in this formulation by $p(y_1, y_2) = 0.5$.

where $\mathrm{KL}\left(\pi\|\pi_{\mathrm{ref}}\right) \doteq \mathbb{E}_{y\sim\pi}\left[\log\frac{\pi(y)}{\pi_{\mathrm{ref}}(y)}\right]$. For the preference objective $J_{\mathrm{P}}$, the $\pi$ and $\pi'$ players optimize their corresponding max-min and min-max objectives, i.e.,

$$\pi_{\mathrm{nash}} \in \arg\max_{\pi\in\Pi}\min_{\pi'\in\Pi} J_{\mathrm{P}}(\pi,\pi',p_{\mathrm{mle}}), \quad \pi'_{\mathrm{nash}} \in \arg\min_{\pi'\in\Pi}\max_{\pi\in\Pi} J_{\mathrm{P}}(\pi,\pi',p_{\mathrm{mle}}). \tag{3}$$

Here, due to the symmetry of the game, a Nash equilibrium exists at the same policy, i.e., $\pi_{\mathrm{nash}} = \pi'_{\mathrm{nash}}$, and the objective can be simplified to optimizing a *single*-player game—termed Self-play Preference Optimization (SPO) in Swamy et al. (2024). We refer to Munos et al. (2023) and Swamy et al. (2024) for formal existence guarantees of this Nash equilibrium.

Alternatively, in the standard *reward*-based setup, given a reward function $r$, the corresponding objective for reward optimization becomes $J_{\mathrm{R}}(\pi,r)$, which is defined as:

$$J_{\mathrm{R}}(\pi,r) \doteq \mathbb{E}_{y\sim\pi}[r(y)] - \beta\mathrm{KL}\left(\pi\|\pi_{\mathrm{ref}}\right). \tag{4}$$

As described earlier, typical reward-based RLHF settings make use of a learned reward $r_{\mathrm{mle}}$ obtained by optimizing (1) for $p_{\mathrm{BTL}}$ in (2), and fine-tune the policy to maximize the expected reward, i.e.,

$$\pi_{\mathrm{rlhf}} \in \arg\max_{\pi\in\Pi} J_{\mathrm{R}}(\pi,r_{\mathrm{mle}}), \quad \text{where} \quad r_{\mathrm{mle}} \in \arg\min_{r} \mathcal{L}_{\mathrm{pref}}\left(p_{\mathrm{BTL}}(r);\mathcal{D}\right). \tag{5}$$

**Pessimism in preference-based policy optimization.** It is a well-understood issue in preference optimization and RLHF that optimizing $J_{\mathrm{P}}(\cdot,\cdot,p_{\mathrm{mle}})$ and $J_{\mathrm{R}}(\cdot,r_{\mathrm{mle}})$ can lead to over-optimization of the corresponding preference and reward functions (Gao et al., 2023; Eisenstein et al., 2024). This behavior arises because $p_{\mathrm{mle}}$ (resp. $r_{\mathrm{mle}}$) has large inaccuracies and/or uncertainties in its predictions outside the support of $\mathcal{D}$, and the policy optimization to maximize the preference (resp. reward), can exploit the such regions with spuriously high scores under $p_{\mathrm{mle}}$ (resp. $r_{\mathrm{mle}}$), resulting in a shift in the distribution of outputs generated by the learned policies ($\pi_{\mathrm{nash}}$ and $\pi_{\mathrm{rlhf}}$). Colloquially, this phenomenon is often termed "*reward hacking*" or "*preference hacking*". Pessimism in both the reward setting (Eisenstein et al., 2024; Liu et al., 2024; Fisch et al., 2024; Cen et al., 2024) and the preference setting (Ye et al., 2024) has been proposed as a way to remedy these issues.

Pessimism in the reward setting leads to a max-min objective, $\pi_{\mathrm{p\text{-}rlhf}} \in \arg\max_{\pi}\min_{r\in\mathcal{R}} J_{\mathrm{R}}(\pi,r)$ where $\mathcal{R}$ is an uncertainty set of reward functions, that is, all reward functions that are consistent with the dataset. Prior works Liu et al. (2024); Fisch et al. (2024) show that for certain choices of $\mathcal{R}$, this game can be solved as regularized policy optimization, by performing the inner optimization in closed form. Note that the use of pessimism here is to combat the uncertainty from a *fixed preference dataset*, even though the policy optimization is done with on-policy samples. In the next section, we extend pessimism to the preference setting (which includes reward-based BTL preferences), and analyze different pessimistic objectives.

## 3 Pessimistic Preference-based RL

We now study extensions of the pessimistic reward-based objective to general preferences. Implementation issues are deferred to §4.

**A pessimistic Nash solution.** In preference-based RL, a pessimistic counterpart of the Nash solution in (3) can be naturally formulated as

$$\pi_{\mathrm{p\text{-}nash}} \in \arg\max_{\pi}\min_{\pi'}\min_{p\in\mathcal{P}} J_{\mathrm{P}}(\pi,\pi',p) \tag{6}$$

where $\mathcal{P} \subseteq \{\mathcal{Y}\times\mathcal{Y}\to[0,1]\}$ defines an uncertainty set over preference functions.[3] In particular we consider the set $\mathcal{P}(\mathcal{D},c) \subset \{\mathcal{Y}\times\mathcal{Y}\to[0,1]\}$, for $c\geq 0$, which is defined as:

$$\{p : \mathcal{L}_{\mathrm{pref}}(p;\mathcal{D}) \leq \mathcal{L}_{\mathrm{pref}}(p_{\mathrm{mle}};\mathcal{D}) + c\}, \tag{7}$$

---

[3]The game not being symmetric leads to $\pi_{\mathrm{p\text{-}nash}} \neq \pi'_{\mathrm{p\text{-}nash}}$.

and choose $\mathcal{P} \doteq \mathcal{P}(\mathcal{D}, c')$ for a value of $c'$, such that $p^\star \in \mathcal{P}(\mathcal{D}, c')$. Note that while $\mathcal{P}$ is defined in terms of the unknown $p^\star$, the set itself can be constructed entirely from data using standard learning-theoretic tools (e.g., likelihood-based confidence regions around $p_{\mathrm{mle}}$), without knowledge of $p^\star$; see Zhu et al. (2023) and Zhan et al. (2024) for concrete constructions. This formulation has been studied previously for certain choices of $\mathcal{P}$ in the tabular (Cui & Du, 2022) and function approximation settings (Ye et al., 2024; Huang et al., 2024a). These works prove that the solution $\pi_{\mathrm{p\text{-}nash}}$ converges to the optimal policy if and only if a condition called *unilateral coverage* holds, which requires that we can compare $\pi_{\mathrm{p\text{-}nash}}$ with any response $y$, within the coverage of the sampling policy $\pi_{\mathrm{data}}$. [4] That is, $p(\pi_{\mathrm{p\text{-}nash}}, y)$ lies within a small interval as we vary $p \in \mathcal{P}$, for all $y$. Here, $\pi_{\mathrm{data}}$ denotes the (unknown) behavior policy that generated the offline preference dataset $\mathcal{D}$; in practice, it can be approximated by supervised fine-tuning on $\mathcal{D}$. This approach has not been empirically evaluated in prior works, as the optimization problem is very challenging with no obvious practical strategies. Before discussing these algorithmic challenges, we will first explore the practical implications of the unilateral coverage requirement.

**The implications of an unconstrained min player.** Consider an illustrative example in Figure 2 which is emblematic of typical RLHF scenarios. There is no context and $\mathcal{Y} = \{y_1, y_2, y_3\}$ and suppose further that we have that $p(y_1, y_2) = 1$ for all $p \in \mathcal{P}$, so we are fully certain about this preference. But we never observe any comparisons involving $y_3$ in our preference data $(\pi_{\mathrm{data}}(y_3) = 0)$, and hence the set $\mathcal{P}$ allows all values $p(y, y_3) \in [0, 1]$ for $y \neq y_3$. To highlight the limitations of pessimism in preference optimization, we consider the problem in absence of regularization, i.e., $\beta = 0$. Then, as illustrated in Figure 2 (Left) and proven in Appendix A, the pessimistic Nash policy $\pi_{\mathrm{p\text{-}nash}}$ satisfies $\pi_{\mathrm{p\text{-}nash}}(y_3) = 0.5$. That is, the policy takes an action out of the support of the sampled dataset w.p. 0.5, where the ground-truth preferences can take completely arbitrary values. Intuitively, this happens because the pessimistic policy has to account for either possibility that $y_1$ is much better than $y_3$, or vice versa. However, in most practical applications, many $y$'s will not be covered in the dataset, even distributionally, and it appears undesirable that the optimal policy obtained by pessimism will then *predominantly* generate such outputs.

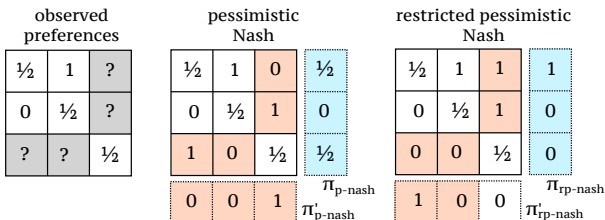

Figure 2: An illustration of the problematic example for pessimistic preference optimization with unrestricted opponents. We assume that $\{y_1, y_2\}$ are well-sampled in the preference data, whereas $y_3$ *is not*—resulting in certain preferences for $y_1$ vs. $y_2$, but completely uncertain preferences for $y_3$ vs. others (**Left**). The $3 \times 3$ matrices above are the optimized pessimistic preference matrices, with the $3d$ vectors the optimized competing policies. Shaded entries represent optimizable variables, and the blue and red shaded entries are the solutions for the max $(\arg\max_\pi)$ and min player $(\arg\min_{\pi'}$ and $\arg\min_{p \in \mathcal{P}})$, resp., see (6). **Middle**: when the opponent $\pi'$ is unrestricted, the optimal policy $\pi_{\mathrm{p\text{-}nash}}$ must hedge and put significant support on $y_3$. **Right**: restricting the support of $\pi'$ to the support of the *preference-data* (i.e., $\{y_1, y_2\}$), avoids this issue, and yields a more reasonable optimal policy $\pi_{\mathrm{rp\text{-}nash}}$.

This example highlights a key distinction between pessimism with rewards versus pessimism with preferences. When using pessimism with rewards, outputs which are not covered in the data tend to get a low score, and a reward-maximizing policy naturally avoids them. But for general preferences, the min player $\pi'$ can choose any output which is not observed in the preference data, and pessimism means that the max player looses to this action in the worst case—forcing the max player to put probability over such an uncovered output as well. We now propose a remedy for this.

**Restricted pessimistic Nash with a constrained min player.** Given the example from Figure 2, an intuitive response is to consider a Nash strategy where the support for the opponent policy $\pi'$ is restricted to actions which are well-sampled in the preference dataset. For tabular scenarios, where we have no contexts and a finite $\mathcal{Y}$, with all possible policies in the class $\Pi$, such a restriction can be carried out by explicitly constraining the support of the min-player in the pessimistic objective (6). However, this does not generalize to more practical scenarios with parametric policies over a large output space. Instead, for such situations,

---

[4]Unilateral coverage requires that for the learned pessimistic policy $\pi_{\mathrm{p\text{-}nash}}$, the preference $p(\pi_{\mathrm{p\text{-}nash}}, y)$ lies within a small interval as we vary $p \in \mathcal{P}$ against competing responses $y$. Informally, $\pi_{\mathrm{data}}$ only needs to provide sufficient comparisons to resolve uncertainty for the specific responses generated by $\pi_{\mathrm{p\text{-}nash}}$, rather than covering the entire response space $\mathcal{Y}$. See Definition B.1 in Appendix B.1 for a formal statement.

we define a subset $\Pi(\pi_{\mathrm{data}}, C) \subseteq \Pi$ to be a set of policies whose outputs are "well-covered" by the sampling policy $\pi_{\mathrm{data}}$, with $C$ denoting a coverage parameter that we define below.

**Definition 3.1** (Covered policy set)**.** *For a given sampling policy $\pi_{\mathrm{data}} \in \Pi$ and constant $C$, the covered policy set $\Pi(\pi_{\mathrm{data}}, C)$ wrt $\pi_{\mathrm{data}}$ is the set of policies such that $\forall \pi, \pi' \in \Pi(\pi_{\mathrm{data}}, C)$ and $\forall p_1, p_2 \in \mathcal{P}$:*
$$\mathbb{E}_{y \sim \pi, y' \sim \pi'}(p_1(y, y') - p_2(y, y'))^2 \leq C \cdot \mathbb{E}_{y, y' \overset{\mathrm{i.i.d.}}{\sim} \pi_{\mathrm{data}}}(p_1(y, y') - p_2(y, y'))^2.$$

What is included in $\Pi(\pi_{\mathrm{data}}, C)$? Clearly, $\pi_{\mathrm{data}} \in \Pi(\pi_{\mathrm{data}}, C)$ when $C \geq 1$. We assume $C \geq 1$ in the sequel to ensure that this containment happens. We can also show (see Appendix B.2 for a short derivation) that $\Pi(\pi_{\mathrm{data}}, C)$ includes the set of all policies with likelihood ratios with respect to $\pi_{\mathrm{data}}$ that are uniformly bounded by $\sqrt{C}$, that is, $\Pi_{\sqrt{C}} \subseteq \Pi(\pi_{\mathrm{data}}, C)$, where $\Pi_{\sqrt{C}} = \{\pi \in \Pi : \|\pi/\pi_{\mathrm{data}}\|_\infty \leq \sqrt{C}\}$. In fact, we further show in Appendix B.3 that when the preference functions are linear in a shared feature map, then this coverage condition is ensured whenever the cross-covariance matrices of $\pi$ and $\pi'$ are sufficiently aligned with the covariance matrix of $\pi_{\mathrm{data}}$.

Perhaps most importantly, however, when defining the following restricted pessimistic Nash solution, $\pi_{\mathrm{rp\text{-}nash}}$, using this notion of coverage, i.e.,

$$\pi_{\mathrm{rp\text{-}nash}} \in \arg\max_{\pi \in \Pi} \min_{\pi' \in \Pi(\pi_{\mathrm{data}}, C)} \min_{p \in \mathcal{P}} J_P(\pi, \pi', p), \tag{8}$$

we can give the following performance guarantee for $\pi_{\mathrm{rp\text{-}nash}}$ under the *ground-truth preference function $p^\star$* which generated the preference dataset $\mathcal{D}$ (which was then used to derive the set $\mathcal{P}$).

**Lemma 3.2** (Preference guarantee for the restricted pessimistic Nash policy)**.** *For the restricted pessimistic Nash policy $\pi_{\mathrm{rp\text{-}nash}}$ (8), and the ground-truth preference function $p^\star$ underlying $\mathcal{D}$, we have for any $\pi \in \Pi(\pi_{\mathrm{data}}, C)$ with $C \geq 1$: $p^\star(\pi_{\mathrm{rp\text{-}nash}}, \pi) \geq 0.5 - 2\sqrt{C}\varepsilon$, where $\varepsilon$ is a bound on how much preference functions in $\mathcal{P}$ can disagree in total variation under $\pi_{\mathrm{data}}$: $\mathbb{E}_{(y, y') \overset{\mathrm{i.i.d.}}{\sim} \pi_{\mathrm{data}}} |p_1(y, y') - p_2(y, y')| \leq \varepsilon, \forall p_1, p_2 \in \mathcal{P}$.*

Appendix B.1 restates Definition 3.1 and Lemma 3.2 with the context $x$ included, along with a proof. Intuitively, Lemma 3.2 is the preference-based analogue of 'single-policy concentrability' guarantees from pessimistic offline RL (Xie et al., 2021; Rashidinejad et al., 2022): the learned policy $\pi_{\mathrm{rp\text{-}nash}}$ is guaranteed to not be dispreferred—up to an error of $2\sqrt{C}\varepsilon$—compared with *any* competing policy whose actions stay within the support of the preference data (i.e., bounded $C$). This result shows that the restricted pessimistic Nash policy is always preferred to all other covered policies in $\Pi(\pi_{\mathrm{data}}, C)$ up to an error term. In comparison, the unrestricted pessimistic Nash solution from (6) does not satisfy this guarantee in general. To see this, consider the example in Figure 2, where the unrestricted pessimistic Nash policy $\pi_{\mathrm{p\text{-}nash}} = [1/2, 0, 1/2]$ is dispreferred to the covered policy $\pi_1 = [1, 0, 0]$ with probability $p^\star(\pi_1, \pi_{\mathrm{p\text{-}nash}}) = 1/2 + \gamma/2$ when $p^\star(y_1, y_3) = \gamma$.

Lemma 3.2 is the natural analog of guarantees from pessimistic offline RL to preference-based RL. In pessimistic offline RL, it is guaranteed that the ground-truth reward of the learned policy is not much worse than that of *any competing policy covered by offline data*. Similarly, Lemma 3.2 guarantees that the learned policy is not dispreferred, up to an error term, compared with *any policy covered by the preference data*, under the ground-truth $p^\star$. Unlike in offline RL, this does require an algorithmic restriction of the min player policy class to the covered policies only, but this restriction only applies to the min policy, while the max policy is not constrained in any way, and can still venture out of support in principle (however, pessimism limits this). Indeed, the proof of Lemma 3.2 crucially relies on both the pessimism over preferences and the restriction of the min player, which is why prior works like Cui & Du (2022) do not enjoy such a guarantee, and instead require a much stronger unilateral coverage assumption.[5] These considerations are particularly pertinent when aligning LLMs with small preference datasets, where the output space is of long sequences over a large vocabulary, of which the preference data only covers a tiny sliver. Developing techniques without explicit min-player restriction that get similar results is an interesting direction for research.

While the restricted pessimistic Nash formulation (8) provides desirable theoretical guarantees, directly enforcing the hard constraint $\pi' \in \Pi(\pi_{\mathrm{data}}, C)$ is intractable for large parametric models. In the next section, we introduce a practical relaxation that replaces the hard membership constraint with a soft KL penalty, enabling efficient optimization.

---

[5]As a reminder, if there is an action $y \in \mathcal{Y}$ such that $\pi_{\mathrm{data}}(y) = 0$, unilateral coverage is violated, but Lemma 3.2 still offers meaningful guarantees against competing policies that do not choose this action.

---

**Algorithm 1** P3O $(\alpha)$

---

**Hyperparameters:** Mixing coefficient $\alpha$, regularization coefficient $\beta$, preference regularization coefficient $\lambda$, EMA parameter $\gamma$, learning rates $(\eta_p, \eta_\pi)$

**Initialize:** $\bar{\pi}_1 = \pi_1 = \pi_{\text{ref}}$, $p_1 = p_{\text{mle}}$

**for** $t = 1, 2, \ldots$ **do**

Set $\pi_{\text{mix}}^\alpha \propto \bar{\pi}_t^{1-\alpha} \pi_{\text{data}}^\alpha$ as mix of $\pi_{\text{data}}$ and the exponential moving avg. $\bar{\pi}_t$ for restricted Nash.

Approximate the current objective (12):

$$J_{\text{P3O}(\alpha)}(\pi_t, p_t) \doteq p_t(\pi_t, \pi_{\text{mix}}^\alpha) - \beta \text{KL}\left(\pi_t \| \pi_{\text{ref}}\right) + \lambda \text{KL}_{\pi_{\text{ref}}}\left(p_{\text{mle}} \| p\right)$$

Update $\pi_{t+1} \leftarrow \pi_t + \eta_\pi \left.\frac{\partial J_{\text{P3O}(\alpha)}(\pi, p_t)}{\partial \pi}\right|_{\pi = \pi_t}$ and $p_{t+1} \leftarrow p_t - \eta_p \left.\frac{\partial J_{\text{P3O}(\alpha)}(\pi_t, p)}{\partial p}\right|_{p = p_t}$

Update $\bar{\pi}_{t+1} \leftarrow \gamma \pi_t + (1 - \gamma)\bar{\pi}_t$

**end for**

---

## 4 P3O: An Efficient Implementation

We now develop an efficient algorithm, Pessimistic Preference-based Policy Optimization (P3O), that approximately solves the restricted pessimistic Nash formulation.

**Approximating the restricted policy set.** A first obstacle to an efficient algorithm is that the definition $\Pi(\pi_{\text{data}}, C)$ is not amenable to easy implementation. However, in KL-regularized preference-based RLHF, there is a natural heuristic to approximate this restriction via an additional KL regularization term. Recall that the central goal of $\Pi(\pi_{\text{data}}, C)$ is to limit optimization to policies $\pi$ which generate responses that are in the coverage of the data generating policy $\pi_{\text{data}}$. We encourage this through adding an additional penalty based on the KL divergence between $\pi'$ and $\pi_{\text{data}}$:

$$\max_\pi \min_{p \in \mathcal{P}} \min_{\pi'} \quad p(\pi, \pi') - \beta \text{KL}\left(\pi \| \pi_{\text{ref}}\right) + (1 - \alpha)\beta \text{KL}\left(\pi' \| \pi_{\text{ref}}\right) + \alpha\beta \text{KL}\left(\pi' \| \pi_{\text{data}}\right). \tag{9}$$

Note that $\pi'$ is regularized with respect to both the reference policy $\pi_{\text{ref}}$ and the sampling policy $\pi_{\text{data}}$, where the added parameter $\alpha$ controls the relative strength of the contribution of $\pi_{\text{ref}}$ versus $\pi_{\text{data}}$. While going from a data-aware constraint in terms of $\Pi(\pi_{\text{data}}, C)$ to KL regularization is lossy, this is for the min player $\pi'$ and only affects the max player $\pi$ through the data-dependent $p$ term.

We use P3O $(\alpha)$ to denote this variant, with P3O being the shorthand for P3O $(0)$. Using a closed-form solution to the inner KL-regularized problem for $\pi'$, we next show how to obtain an equivalent, but greatly simplified, objective for $\pi$. First, we define shorthand $\pi_{\text{mix}}^\alpha(y; \pi_1, \pi_2)$ as: $\pi_{\text{mix}}^\alpha(y; \pi_1, \pi_2) \propto \pi_1^{1-\alpha}(y)\pi_2^\alpha(y)$, which is implemented via convex combinations of logits (Appendix E). Optimizing $\pi$ against an appropriate mixed distribution is then equivalent to solving for (9), as we show below.

**Lemma 4.1.** *The optimization problem in* (9) *is equivalent to the following objective, assuming that the minimization over $\pi'$ is over all possible policies:*

$$\max_\pi \min_{p \in \mathcal{P}} -\log \mathbb{E}_{y \sim \pi_{\text{mix}}^\alpha(\pi_{\text{ref}}, \pi_{\text{data}})}\left[\exp(-p(\pi, y)/\beta)\right] - \beta \text{KL}\left(\pi \| \pi_{\text{ref}}\right). \tag{10}$$

We prove this equivalence in Appendix C. We also note that replacing $\pi_{\text{mix}}^\alpha(\pi_{\text{ref}}, \pi_{\text{data}})$ with $\pi_{\text{ref}}$ (i.e., $\alpha = 0$) gives a reformulation for the pessimistic Nash objective with no support restrictions (6).

**Approximating the log-partition function.** The objective in Lemma 4.1 simplifies the inner minimization to a single variable, but at the cost of changing the objective to have a more complicated log-partition function term. Consequently, we can no longer get unbiased stochastic gradients of the objective from a mini-batch of data, due to the non-linearity of the log outside of the expectation.

To derive a practical algorithm, we leverage ideas from variational inference (Jordan et al., 1999) to approximate the log-partition function. This yields the following result, proved in Appendix D.

**Lemma 4.2.** *For any policies $\pi, \bar{\pi} \in \Pi$, there is a constant $\kappa$ independent of $\pi$ and $p$, such that:*

$$\min_{p \in \mathcal{P}} -\log \mathbb{E}_{y \sim \pi^\alpha_{\text{mix}}(\pi_{\text{ref}}, \pi_{\text{data}})} \left[\exp(-p(\pi, y)/\beta)\right] \leq \min_{p \in \mathcal{P}} \mathbb{E}_{y \sim \pi^\alpha_{\text{mix}}(\bar{\pi}, \pi_{\text{data}})} \left[p(\pi, y)/\beta\right] + \kappa, \tag{11}$$

Due to the direction of the inequality, Lemma 4.2 gives only an *upper bound* for our objective in Lemma 4.1, and therefore maximizing the two is not equivalent. Nevertheless, the approximate objective is tractable, and simply takes the form of optimizing preferences against some comparator $\bar{\pi}$ mixed with the sampling distribution. Furthermore, the approximate objective at any fixed value of $p$ is tight when $\bar{\pi}(y)^{1-\alpha} \propto \exp(-p(\pi, y)/\beta)$, which resembles the multiplicative weight updates observed in prior self-play algorithms (Swamy et al., 2024; Munos et al., 2023). Since the current preference function iterate $p_t$ is slowly moving during gradient descent, with this motivation in hand, we choose the competitor policy $\bar{\pi}$ to be an exponentially moving average of past policy iterates in our experiments, giving our algorithm a pessimistic self-play flavor.

**Approximating the preference uncertainty set.** As a final step, we replace the constrained optimization over the preference uncertainty set $\mathcal{P}$ with an unconstrained optimization over all preference functions in some parametric family by adding an additional loss term $-\mathcal{L}_{\text{pref}}(p; \mathcal{D})$ (Liu et al., 2024), corresponding to the Lagrangian form of the constraint defining $\mathcal{P}$. The objective function then becomes $\mathbb{E}_{y \sim \pi^\alpha_{\text{mix}}(\bar{\pi}, \pi_{\text{data}})}[p(\pi, y)] - \beta \text{KL}(\pi \| \pi_{\text{ref}}) + \lambda \mathcal{L}_{\text{pref}}(p; \mathcal{D})$, where we rescaled the objective to absorb the $1/\beta$ on $p$ into the corresponding hyper-parameters of the KL and likelihood loss components (i.e., $\beta, \lambda$). The above objective requires us to also load the preference dataset $\mathcal{D}$ while trying to learn the policy, which can be somewhat inconvenient. To circumvent this issue, however, we can instead simply regularize the preference model to stay close to $p_{\text{mle}}$, i.e.,

$$J_{\text{P3O}(\alpha)}(\pi, p) \doteq p\left(\pi, \pi^\alpha_{\text{mix}}(\bar{\pi}, \pi_{\text{data}})\right) - \beta \text{KL}(\pi \| \pi_{\text{ref}}) + \lambda \text{KL}_{\pi_{\text{data}}}(p_{\text{mle}} \| p), \tag{12}$$

where $\text{KL}_{\pi_{\text{data}}}(p_{\text{mle}} \| p)$ is defined as $\mathbb{E}_{y, y' \overset{\text{i.i.d.}}{\sim} \pi_{\text{data}}}[\text{KL}(p_{\text{mle}}(y, y') \| p(y, y'))]$. If the MLE solution $p_{\text{mle}}$ is a good approximation to the true preferences $p^\star$ on $y, y' \overset{\text{i.i.d.}}{\sim} \pi_{\text{data}}$, then this KL divergence provides a good approximation to the likelihood-based version. See Algorithm 1 for the pseudocode.

Having introduced P3O, which handles general preferences, we can extend it to the special case where preferences are parameterized by a reward function under the BTL model. In this setting, we replace the general preference function $p$ with $p_{\text{BTL}}(r)$ in (12), resulting in the following objective:

$$J_{\text{PRPO}(\alpha)}(\pi, r) \doteq p_{\text{BTL}}\left(\pi, \pi^\alpha_{\text{mix}}(\bar{\pi}, \pi_{\text{data}}); r\right) - \beta \text{KL}(\pi \| \pi_{\text{ref}}) + \lambda \text{KL}_{\pi_{\text{data}}}(p_{\text{BTL}}(r_{\text{mle}}) \| p_{\text{BTL}}(r)). \tag{13}$$

We refer to the algorithm that optimizes (13) as Pessimistic Reward-based Policy Optimization (PRPO). Appendix E provides the pseudo code and further discussion on PRPO.

**Quality of the approximations.** The derivation of P3O involves several approximations—replacing the hard constraint on $\pi'$ with a soft KL penalty, and upper-bounding the log-partition function via a variational argument. To empirically assess their impact, we conduct experiments in a tabular setting (Appendix F) where we compare the exact restricted pessimistic Nash objective (EP3O) with its approximate counterpart (P3O) using brute-force search. As shown in Figures 4 and 5 therein, the approximate objective closely tracks the exact one, and—crucially—preserves the core property of mitigating over-optimization while remaining in support of $\pi_{\text{data}}$.

## 5 Experimental Results

In this section, we empirically study three questions that probe how different strategies, ordered by complexity, can mitigate preference-hacking. We examine these questions on two benchmarks: the TL;DR summarization task and the Anthropic helpfulness task. Both these tasks have been studied in prior works on RLHF (Munos et al., 2023; Calandriello et al., 2024) and reward hacking (Eisenstein et al., 2024; Fisch et al., 2024), are known to exhibit these behaviors numerically, and the qualitative hacking behaviors are well understood, allowing for

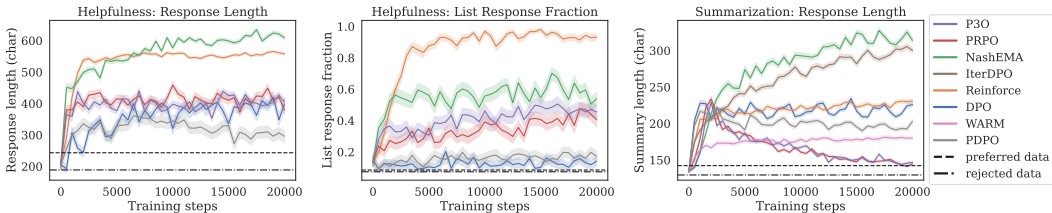

Figure 3: **Qualitative results for the helpfulness and summarization tasks:** On helpfulness, both response length and list formats are common reward hacks Eisenstein et al. (2024). While all policies do generate longer responses than $\pi_{\text{data}}$, REINFORCE and NashEMA converge on generations that are $\approx 40 - 50\%$ longer than those of P3O and PRPO (**left**). Similarly, REINFORCE degenerates to responses that are nearly all formatted as lists with NashEMA at over 50% and P3O and PRPO share closer to 40% (**middle**). On summarization, DPO, REINFORCE, IterDPO and NashEMA all show clear signs of length hacking—also a pervasive issue on this task Eisenstein et al. (2024); Singhal et al. (2023); Park et al. (2024). WARM and PDPO reduce length hacking at the cost of worse win-rates. Both PRPO and P3O converge to the average length of preferred responses in $\pi_{\text{data}}$, all while also achieving the highest win-rates (**right**).

| Method | Preference / reward model | Mechanism | Summarization | | | | Helpfulness | | | | |
|---|---|---|---|---|---|---|---|---|---|---|---|
| | | | Win-rate ↗ vs. | | | Hacking ↘ | Win-rate ↗ vs. | | | Hacking ↘ | |
| | | | $\pi_{\text{ref}}$ | P3O | PRPO | Length | $\pi_{\text{ref}}$ | P3O | PRPO | Length | List |
| P3O | Preference | Pessimism | 0.91 | 0.50 | 0.57 | 1.03x | 0.89 | 0.50 | 0.59 | 1.81x | 5.98x |
| PRPO | Reward | Pessimism | 0.88 | 0.43 | 0.50 | 1.09x | 0.86 | 0.41 | 0.50 | 1.88x | 5.61x |
| NashEMA | Preference | KL | 0.83 | 0.31 | 0.40 | 2.17x | 0.90 | 0.67 | 0.76 | 2.61x | 8.29x |
| IterDPO | Preference | KL | 0.89 | 0.44 | 0.52 | 2.04x | - | - | - | - | - |
| REINFORCE | Reward | KL | 0.84 | 0.35 | 0.43 | 1.62x | 0.78 | 0.46 | 0.54 | 2.32x | 11.59x |
| DPO | - | KL | 0.82 | 0.34 | 0.41 | 1.47x | 0.73 | 0.31 | 0.38 | 1.73x | 2.40x |
| WARM | Reward | Model Avg. | 0.74 | 0.26 | 0.31 | 1.27x | - | - | - | - | - |
| PDPO | - | Pessimism | 0.83 | 0.35 | 0.42 | 1.41x | 0.71 | 0.25 | 0.31 | 1.48x | 2.33x |

Table 1: Summary of our experimental evaluation, comparing our methods (P3O and PRPO) against several baselines on summarization and "helpful assistant" tasks. We evaluate each method by the win-rates of the best checkpoints, judged by Gemini 1.5 Flash, as well as response length and list frequency, quantifying known preference hacking behavior on these tasks. Leveraging pessimism, our method consistently achieve high win-rates and good preference hacking metrics, strictly beating baselines on summarization. NashEMA (helpfulness) and IterDPO (summarization) obtain high win-rates, but generate much longer responses. When looking for both helpful *and* concise responses, NashEMA's win-rate against P3O and PRPO drops dramatically (see Figure 11), demonstrating their robustness. CIs are available in Table 2 in the appendix.

a robust evaluation of our methods. We first describe the tasks, experimental setup, and baselines, then report results addressing each research question. We rely on the approximate pessimistic objective $J_{\text{P3O}(\alpha)}$ (12) for our proposed methods, P3O and PRPO, to analyze how pessimism influences preference-hacking behavior. For these experiments, we set $\alpha = 0$, as $\pi_{\text{ref}}$ and $\pi_{\text{data}}$ are similar, meaning $\alpha$ does not significantly influence the behavior of P3O in this context.

**Task 1: Summarization.** To demonstrate the effectiveness of our approach in mitigating preference hacking, we compare it against existing preference optimization methods on the popular TL;DR summarization task (Völske et al., 2017; Stiennon et al., 2020). Following prior studies on reward hacking for TL;DR (Eisenstein et al., 2024), we train an MLE preference model $p_{\text{mle}}$ as well as MLE reward model $r_{\text{mle}}$ by fine-tuning a T5 XL (3B) model (Raffel et al., 2020; Roberts et al., 2023) on the preference dataset. The initial policy $\pi_{\text{ref}}$ is obtained by supervised fine-tuning a T5 large model (770M) on the human reference summaries in TL;DR. Choosing a larger preference model than the policy is a commonly employed strategy for mitigating hacking (Eisenstein et al., 2024). We initialize the training preference model $p_1 \doteq p_{\text{mle}}$ in case of P3O, and $r_1 \doteq r_{\text{mle}}$ in case of PRPO.

**Task 2: Helpfulness.** We evaluate our method at larger scale on the Anthropic Helpfulness task (Bai et al., 2022) using 8B PaLM-based (Anil et al., 2023) policy, reward, and preference models. The task consists of human–assistant dialogues, where the goal is to generate a helpful and engaging next-turn response. We train MLE preference and reward models by fine-tuning a pretrained PaLM model on the preference data; these

checkpoints also initialize the preference and reward models for P3O/PRPO. The reference policy $\pi_{\text{ref}}$ is an instruction-tuned PaLM model.

**Baselines.** We benchmark our two pessimistic algorithms—P3O and PRPO —against a representative set of RLHF methods that covers predominant algorithmic paradigms:

- **Non-pessimistic baselines:** DPO (an offline method) (Rafailov et al., 2023), REINFORCE (shown to outperform PPO on these tasks (Ahmadian et al., 2024)) (Williams, 1992), and the preference-model variant NashEMA (Munos et al., 2023).
- **Pessimistic or hybrid baselines (TL;DR only):** PDPO, a pessimistic version of DPO (Fisch et al., 2024); Iterative-DPO, an online version of DPO; and WARM, a model-averaging approach (Ramé et al., 2024).

**Evaluation protocol.** We use a prompted Gemini-1.5 Flash (Gemini Team, 2024) model to compare the responses of each policy to those of $\pi_{\text{ref}}$ on the evaluation prompts. To capture qualitative failure modes, we also track two well-documented forms of reward hacking: (i) *length hacking* in TL;DR and helpfulness, where models gain reward by writing overly verbose summaries, and (ii) *style hacking* in helpfulness, where answers degenerate into exhaustive bullet-point lists (Eisenstein et al., 2024; Singhal et al., 2023; Park et al., 2024). These metrics let us judge not only who wins preferences, but also *how* they win. This two-pronged strategy is deliberate: a frontier LLM judge is standard in recent literature (Eisenstein et al., 2024; Wu et al., 2024; Zheng et al., 2023) and is substantially larger than our models, but we supplement it with evaluator-independent distributional metrics to verify that policies are not merely gaming the evaluation.

**Question 1: Can careful hyperparameter tuning of standard RLHF methods—especially the KL penalty—keep preference hacking in check without hindering learning?** Figure 1 shows that standard RLHF methods—DPO and REINFORCE—plateau at a noticeably lower preference score, with DPO even dropping after extended training. Ablations in Appendix G confirm that both methods need a *large* KL penalty to prevent overoptimization; such heavy regularization, however, leaves the policy little room to improve further. Even in this setting, the qualitative plots in Figure 3 reveal clear "length-hacking" on both tasks and list-style inflation in helpfulness, especially for REINFORCE. In short, aggressive tuning dampens—but does not eliminate—preference hacking, and it caps the attainable performance. The two exceptions are NashEMA (helpfulness) and IterDPO (summarization), but both still suffer from severe length hacking as shown in Table 1 and Figure 3.

**Question 2: Can straightforward tactics such as model ensembling / averaging reduce preference hacking?** Figure 1 plots the behavior of the model-averaging baseline WARM for TL;DR. Averaging does reduce the length-hacking seen in REINFORCE—the summaries stay noticeably closer to the reference length (Table 1)—but they are *still* longer than the reference summaries, and the preference score plateaus far below that of the P3O. This echoes the conclusion of Eisenstein et al. (2024), who found that ensembling reduces hacking symptoms without removing them altogether.

**Question 3: Does a pessimistic objective offer a stronger approach, particularly when compared to its non-pessimistic counterparts, and how do its benefits hold up under alternative evaluation metrics?** Figure 1 shows that our new pessimistic objectives deliver the highest preference scores: both P3O and PRPO outperform against the non-pessimistic baselines, including NashEMA, which shares the training using general preference models but omits pessimism. Prior work on pessimism (Fisch et al., 2024; Cen et al., 2024; Liu et al., 2024) in RLHF, as represented by PDPO in Figure 1 and Table 1, ends up overregularizing and underoptimizing, however. This mirrors the empirical results in Fisch et al. (2024), and might be due to the implicit minimization over reward models in their work, as opposed to an explicit min-max optimization in this work. The pairwise comparisons in Table 1 further confirm the trend—P3O and PRPO win head-to-head against every competitor. The lone outlier is NashEMA on *helpfulness*, where Gemini-1.5 Flash tends to favour NashEMA's longer, list-heavy answers. When we switch to an evaluation that also rewards conciseness (Figure 11), P3O and PRPO again take the lead. Between the two pessimistic methods, the preference-based P3O edges out the reward-based PRPO, highlighting the benefit of modelling richer feedback.

Figure 3 illustrates why; non-pessimistic methods inflate summary length in TL;DR and default to long bullet lists in helpfulness, while P3O and PRPO stay closer to the reference characteristics. We also see that this is a learned behavior in that even P3O and PRPO start inflating the length of responses at first, but the pessimistic preferences pull the response length back over time. Interestingly, Figure 10 shows that REINFORCE maintains a lower KL divergence to $\pi_{\text{ref}}$ than the pessimistic methods, yet still hacks the reward; P3O and PRPO move farther in distribution space but along dimensions that improve perceived quality. On helpfulness, responses from REINFORCE and NashEMA grow to roughly $1.5\times$ the length produced by P3O and PRPO, and nearly every REINFORCE answer appears as a list—patterns absent from the pessimistic outputs.

**Practical considerations.** Pessimism significantly increases the memory footprint required to train these models because it keeps an extra policy copy and a reward (or preference) model. To make this feasible for larger models, Appendix G.3 reports a variant that trains only $\sim 14\%$ of the reward-model parameters; it matches full-parameter performance while cutting memory use sharply.

In Appendix G.2, we present further results on the training dynamics, which indicate a steadily improved objective for all the methods, even when the evaluation performance is non-monotonic. We also show some cherry-picked responses in Appendix I for both summarization and helpfulness tasks to illustrate the stylistic differences in their responses. We refer the reader to Appendix G for these and further details and results of our empirical evaluation. We also present a study the benefits of using restricted pessimism in simple tabular settings with exact optimization in Appendix F.

## 6 Literature Review

Offline RL is primarily concerned with learning a policy from a fixed dataset, a problem that has attracted considerable attention. Many works focus on scenarios with sufficiently broad dataset coverage (Antos et al., 2007; 2008; Munos, 2003; Munos & Szepesvári, 2008; Farahmand et al., 2010; Chen & Jiang, 2019; Xie & Jiang, 2020), though such assumptions tend to be overly restrictive and seldom hold in real-world situations. Consequently, recent research has shifted toward the more realistic setting of inadequate coverage (Wang et al., 2020), aiming to learn a "best effort" policy (Liu et al., 2020). Two major strategies have emerged to handle poor coverage: behavior policy regularization (Fujimoto et al., 2019; Laroche et al., 2019; Kumar et al., 2019; Wu et al., 2019; Jaques et al., 2019; Kostrikov et al., 2021; Xiao et al., 2023) and pessimism in the face of uncertainty (Kumar et al., 2020; Liu et al., 2020; Kidambi et al., 2020; Yu et al., 2020; Buckman et al., 2020; Jin et al., 2021; Zanette et al., 2021; Xie et al., 2021; Cheng et al., 2022; Zhang et al., 2024; Koppel et al., 2024). In limited-data regimes, *pessimism* has been shown to provide strong theoretical guarantees for the resulting policy (Buckman et al., 2020; Jin et al., 2021), achieving min–max optimality in linear MDPs. Moreover, it has been successfully incorporated into both linear (Zanette et al., 2021) and deep RL (DRL) settings (Xie et al., 2021; Cheng et al., 2022).

**Pessimism and robustness in RLHF via reward modeling.** Behavior policy regularization has also been explored in language models (Jaques et al., 2019), alongside standard RLHF approaches that commonly regularize to a reference policy (Stiennon et al., 2020). When the reward function is learned from limited data, inaccuracies naturally arise, mirroring the challenge in value-based offline RL where value estimates become unreliable in underrepresented state-action regions. Pessimism thus serves as a compelling remedy and has recently been investigated in the *reward-based* setting. Fisch et al. (2024); Liu et al. (2024) and related works consider uncertainty-aware/robustification schemes (including distillation and implicit adversarial regularization). Cen et al. (2024) propose value-regularized preference optimization (VPO), which modulates optimism/pessimism on the reward side. Concurrently, Eisenstein et al. (2024); Coste et al. (2023) study reward-model ensembles and show they mitigate—but do not eliminate—over-optimization. More recent work learns a *pessimistic reward model* so that policy optimization can proceed without explicit KL penalties yet avoid reward hacking on summarization benchmarks (Xu et al., 2025). Complementary efforts improve reward-model *robustness* directly, e.g., via batch-wise normalization/constraints that reduce verbosity and length bias while *increasing* preference win-rates when used in RLHF (Hong et al., 2025). There is also parallel progress on discouraging OOD exploitation during policy optimization itself, such as behavior-supported regularization (Dai et al., 2025) and importance-sampling based corrections for direct alignment

algorithms (Nguyen et al., 2025). Finally, the *energy loss* perspective offers a diagnostic and a mitigation for reward-hacking phenomena observed during RLHF (Miao et al., 2025). Separately, $\chi^2$-regularization (Huang et al., 2024b) offers a principled offline penalty but does not account for reward-model uncertainty and is limited to the BTL setting.

**General preference learning and self-play/Nash formulations.** On the preference-learning front, recent work has relaxed the BTL assumption, either by bypassing the need for a separate scalar reward model (Azar et al., 2023; Tang et al., 2024) or adopting self-play/Nash-style approaches (Munos et al., 2023; Swamy et al., 2024; Calandriello et al., 2024; Rosset et al., 2024; Wu et al., 2024; Zhang et al., 2025). These methods learn from *general* preferences and can yield strong practical algorithms. However, they typically do not address *preference hacking* under limited data coverage. Ye et al. (2024) introduce a *pessimistic preference objective* coupled with an exploration scheme, but in an *unrestricted* minimax setting; they do not provide a coverage-aware restriction on the minimizer nor a practical surrogate tailored for general preferences. In contrast, our work formulates a *restricted* pessimistic Nash solution with explicit coverage-aware constraints on the min player and derives an efficient relaxation (P3O/PRPO) that incorporates *pessimism directly into general preference optimization*, aiming to prevent preference hacking without relying on heavy KL penalties.

**Exploration, regularization, and recent theory.** Recent work clarifies when standard regularizers or simple sampling suffice for efficiency. For online, preference-driven training, information-directed sampling yields regret guarantees, and KL-regularized RL admits logarithmic-regret analyses that explain KL's role in shaping the optimization landscape (Qi et al., 2025; Zhao et al., 2025). Separately, greedy sampling against empirical estimates can be provably efficient for RLHF under general preferences with KL-regularized targets (Wu et al., 2025). At the empirical–theoretical interface, design analyses of offline preference methods (e.g., DPO/IPO/SLiC) document instability and collapse under certain losses, motivating objective-level remedies (Agarwal et al., 2025). Orthogonal to our focus, COV provides length-aware bounds and regularizers that jointly address corrupted feedback, over-optimization, and verbosity, and shows an equivalence between DPO-COV and RLHF-COV (Chen et al., 2025). These lines of work are complementary to our contribution: they inform exploration and loss/regularization choices, whereas we study *pessimism inside general preference optimization* with coverage-aware restrictions and practical solvers.

# 7   Conclusion

Modern RLHF methods suffer from a significant tendency to overoptimize spurious preferences (or rewards) that are derived from faulty preference (or reward) models. In this work, we introduced pessimistic, preference-based RLHF objectives, which carefully balance uncertainty with effective learning. In particular, we theoretically analyzed the limitations of existing pessimistic estimators, and derive a novel formulation for a restricted, pessimistic Nash solution with provable advantages. Empirical results on multiple tasks and models demonstrate that our approach effectively resists overoptimization while outperforming standard RLHF baselines—highlighting the potential of pessimistic objectives for achieving robust language model alignment.

**Broader Impact.** Reward and preference hacking is a safety concern in the deployment of RLHF-trained language models: a policy may produce outputs that appear favorable under a flawed evaluator—one trained on limited data, with no exposure to the distribution of responses generated by the learned policy—while being unhelpful or potentially harmful. By developing objectives that are provably robust to such over-optimization, our work contributes to safer and more reliable AI alignment.

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

# Contents

# A    Optimal Solution of the Example in Figure 2

Let $\pi_i = \pi(y_i)$ and $p_{23} = p(y_2, y_3)$ and $p_{13} = p(y_1, y_3)$. Then we can write the objective in this example as

$$V^\star = \max_\pi \min_{\pi'} \min_{p \in \mathcal{P}} J(\pi, \pi', p)$$

$$\text{where } J(\pi, \pi', p) = p(\pi, \pi') = \big(\; 0.5\pi_1\pi_1' + 1 \cdot \pi_1\pi_2' + p_{13}\pi_1\pi_3'$$
$$+ 0 \cdot \pi_2\pi_1' + 0.5\pi_2\pi_2' + p_{23}\pi_2\pi_3'$$
$$+ (1 - p_{13})\pi_3\pi_1' + (1 - p_{23})\pi_3\pi_2' + 0.5\pi_3\pi_3' \big).$$

**Upper-bound on $V^\star$:**    First note that for any $\pi$:

$$\min_{\pi'} \min_{p \in \mathcal{P}} J(\pi, \pi', p) \leq 0.5\pi_3 \qquad \text{and} \qquad \min_{\pi'} \min_{p \in \mathcal{P}} J(\pi, \pi', p) \leq 0.5\pi_1.$$

The first inequality follows by considering the choice $p_{13} = p_{23} = 1$, and the second inequality from considering the choice $\pi_1' = 1$ and $p_{13} = 1$. Since both bounds hold simultaneously and $\pi_1 + \pi_3 \leq 1$, we can conclude that

$$V^\star \leq 0.25.$$

**Lower-bound on $V^\star$:**    Choosing $\pi_1 = \pi_3 = 0.5$, we see that the objective value can be written as

$$J(\pi, \pi', p) = \big(0.25\pi_1' + 0.5\pi_2' + 0.5p_{13}\pi_3'$$
$$+ 0.5(1 - p_{13})\pi_1' + 0.5(1 - p_{23})\pi_2' + 0.25\pi_3'\big).$$

First we observe that the minimum of this quantity is always attained at $p_{23} = 1$ and thus we can ignore the penultimate term. Consider now two cases:

- Case $\pi_1' \leq \pi_3'$: Then the coefficient of $p_{13}$ is non-negative and the minimum is attained at $p_{13} = 0$. This allows us to simplify the expression further as

$$\min_{\pi'} \min_{p \in \mathcal{P}} J(\pi, \pi', p) = \min_{\pi'} 0.25\pi_1' + 0.5\pi_2' + 0.5\pi_1' + 0.5\pi_2' + 0.25\pi_3'$$
$$= \min_{\pi'} 0.75\pi_1' + \pi_2' + 0.25\pi_3'$$
$$= 0.25$$

  where we choose $\pi_3' = 1$ in the last step.

- Case $\pi_1' \geq \pi_3'$: Then the coefficient of $p_{13}$ is non-positive and the minimum is attained at $p_{13} = 1$. This gives

$$\min_{\pi'} \min_{p \in \mathcal{P}} J(\pi, \pi', p) = \min_{\pi'} 0.25\pi_1' + 0.5\pi_2' + 0.5\pi_3' + 0.5\pi_2' + 0.25\pi_3'$$
$$= \min_{\pi'} \min_{P \in \mathcal{P}} 0.25\pi_1' + \pi_2' + 0.75\pi_3'$$
$$= 0.25$$

  where the optimal solution is to choose $\pi_1' = 1$.

Combining both cases, we can conclude that

$$V^\star \geq 0.25.$$

**Optimal solution.**    Combining both upper- and lower-bounds, we can conclude that $V^\star = 0.25$ which is attained at $\pi_1 = \pi_3 = 0.5$.

# B  Definition and Analysis of Restricted Nash Policy

## B.1  Restatement and proof of Lemma 3.2

We restate Definition 3.1 and Lemma 3.2 over here, but with conversation context ($x$) included in the equations for clarity, hence for this section we redefine $\mathcal{P} \subseteq \{\mathcal{X} \times \mathcal{Y} \times \mathcal{Y} \to [0,1]\}$. We also use $x \sim \mathcal{D}$ to denote a context $x$ sampled from the offline dataset.

**Definition B.1** (Unilateral coverage (Adapted from Cui & Du (2022)))**.** *Given an uncertainty set $\mathcal{P}$ and a sampling policy $\pi_{\mathrm{data}}$, a target policy $\pi$ satisfies $\delta$-unilateral coverage with respect to $\pi_{\mathrm{data}}$ and $\mathcal{P}$ if for all responses $y \in \mathcal{Y}$:*

$$\sup_{p_1, p_2 \in \mathcal{P}} |p_1(\pi, y) - p_2(\pi, y)| \leq \delta.$$

*That is, the preference data generated by $\pi_{\mathrm{data}}$ must be sufficient to resolve uncertainty in the preference of $\pi$ against* every *possible response $y \in \mathcal{Y}$—including responses outside the support of $\pi_{\mathrm{data}}$. This is a strong requirement: in the LLM setting where $\mathcal{Y}$ is the space of all possible token sequences, no finite preference dataset can provide coverage for all $y$, making this condition effectively impossible to satisfy. This motivates the restricted pessimistic Nash formulation in Definition 3.1, which instead constrains the comparator $\pi'$ to the covered policy set.*

**Definition B.2** (Covered policy set)**.** *For a given sampling policy $\pi_{\mathrm{data}} \in \Pi$ and constant $C$, the covered policy set $\Pi(\pi_{\mathrm{data}}, C)$ with respect to $\pi_{\mathrm{data}}$ is the set of policies such that $\forall \pi, \pi' \in \Pi(\pi_{\mathrm{data}}, C)$ and $\forall p_1, p_2 \in \mathcal{P}$,*

$$
\begin{aligned}
& \mathbb{E}_{x\sim\mathcal{D}, y\sim\pi(\cdot|x), y'\sim\pi'(\cdot|x)}(p_1(x,y,y') - p_2(x,y,y'))^2 \\
& \leq C \cdot \mathbb{E}_{x\sim\mathcal{D}, y, y' \overset{\mathrm{i.i.d.}}{\sim} \pi_{\mathrm{data}}(\cdot|x)}(p_1(x,y,y') - p_2(x,y,y'))^2.
\end{aligned}
\tag{14}
$$

**Lemma B.3** (Preference guarantee for the restricted pessimistic Nash policy)**.** *We denote the restricted pessimistic Nash policy by $\pi_{\mathrm{rp\text{-}nash}}$ from Eq. (8), and let $p^\star$ be the ground-truth preference function underlying $\mathcal{D}$. Then we have that for any $\pi \in \Pi(\pi_{\mathrm{data}}, C)$ with $C \geq 1$:*

$$p^\star(\pi_{\mathrm{rp\text{-}nash}}, \pi) \geq \frac{1}{2} - 2\sqrt{C}\varepsilon,$$

*where $\varepsilon$ is a bound on how much preference functions in $\mathcal{P}$ can disagree in total variation under $\pi_{\mathrm{data}}$: $\mathbb{E}_{x\sim\mathcal{D}, (y,y') \overset{\mathrm{i.i.d.}}{\sim} \pi_{\mathrm{data}}(\cdot|x)}|p_1(x,y,y') - p_2(x,y,y')| \leq \varepsilon, \ \forall p_1, p_2 \in \mathcal{P}.$*

*Proof of Lemma 3.2.* Let $\pi^\star_{\mathrm{r\text{-}nash}}$ be a restricted Nash solution under the true preference function $p^\star$:

$$
\begin{aligned}
\pi^\star_{\mathrm{r\text{-}nash}} &= \underset{\pi \in \Pi(\pi_{\mathrm{data}}, C)}{\arg\max} \ \underset{\pi' \in \Pi(\pi_{\mathrm{data}}, C)}{\min} \mathbb{E}_{x\sim\mathcal{D}, y\sim\pi(\cdot|x), y'\sim\pi'(\cdot|x)}[p^\star(x,y,y')] \\
&= \underset{\pi \in \Pi(\pi_{\mathrm{data}}, C)}{\arg\max} \ \underset{\pi' \in \Pi(\pi_{\mathrm{data}}, C)}{\min} p^\star(\pi, \pi').
\end{aligned}
$$

We start by noting that $\pi^\star_{\mathrm{r\text{-}nash}}$ is solving an anti-symmetric two player zero-sum game and the constraint set $\Pi(\pi_{\mathrm{data}}, C)$ is a convex set whenever $\Pi$ is convex. To see this, consider two policies $\pi, \pi' \in \Pi(\pi_{\mathrm{data}}, C)$ and $\alpha \in [0,1]$. Then for any $\pi'' \in \Pi(\pi_{\mathrm{data}}, C)$ and $p \in \mathcal{P}$, we have

$$
\begin{aligned}
& \mathbb{E}_{x\sim\mathcal{D}, y\sim(\alpha\pi(\cdot|x)+(1-\alpha)\pi'(\cdot|x)), y'\sim\pi''(\cdot|x)}(p_1(x,y,y') - p_2(x,y,y'))^2 \\
& = \alpha\mathbb{E}_{x\sim\mathcal{D}, y\sim\pi(\cdot|x), y'\sim\pi''(\cdot|x)}(p_1(x,y,y') - p_2(x,y,y'))^2 \\
& \quad + (1-\alpha)\mathbb{E}_{x\sim\mathcal{D}, y\sim\pi'(\cdot|x), y'\sim\pi''(\cdot|x)}(p_1(x,y,y') - p_2(x,y,y'))^2 \\
& \leq C \cdot \mathbb{E}_{x\sim\mathcal{D}, y, y' \overset{\mathrm{i.i.d.}}{\sim} \pi_{\mathrm{data}}(\cdot|x)}(p_1(x,y,y') - p_2(x,y,y'))^2
\end{aligned}
$$

where the first inequality uses linearity of expectation and the second that $\pi, \pi' \in \Pi(\pi_{\mathrm{data}}, C)$. Thus, $\alpha\pi + (1-\alpha)\pi' \in \Pi(\pi_{\mathrm{data}}, C)$ and $\Pi(\pi_{\mathrm{data}}, C)$ is convex. As a consequence of this, we

have that $\pi^\star_{\text{r-nash}} \in \arg\min_{\pi \in \Pi(\pi_{\text{data}}, C)} p^\star(\pi^\star_{\text{r-nash}}, \pi)$ and $p^\star(\pi^\star_{\text{r-nash}}, \pi^\star_{\text{r-nash}}) = 0.5$. Let $\pi'_{\text{rp-nash}} \in \arg\min_{\pi \in \Pi(\pi_{\text{data}}, C)} \min_{p \in \mathcal{P}} p(\pi_{\text{rp-nash}}, \pi)$. Then we have by definition:

$$
\begin{aligned}
p^\star(\pi_{\text{rp-nash}}, \pi) =& p^\star(\pi_{\text{rp-nash}}, \pi) - p^\star(\pi^\star_{\text{r-nash}}, \pi^\star_{\text{r-nash}}) + 0.5 \\
\geq& \min_{p \in \mathcal{P}} p(\pi_{\text{rp-nash}}, \pi'_{\text{rp-nash}}) - p^\star(\pi^\star_{\text{r-nash}}, \pi^\star_{\text{r-nash}}) + 0.5 \\
\geq& \min_{p \in \mathcal{P}} \min_{\pi' \in \Pi(\pi_{\text{data}}, C)} p(\pi^\star_{\text{r-nash}}, \pi') - p^\star(\pi^\star_{\text{r-nash}}, \pi^\star_{\text{r-nash}}) + 0.5,
\end{aligned}
$$

where the first inequality is due to the definition of $\pi'_{\text{rp-nash}}$, and the second follows from the definition of $\hat{\pi}$. Let $\widetilde{\pi} \in \arg\min_{\pi' \in \Pi(\pi_{\text{data}}, C)} \min_{p \in \mathcal{P}} p(\pi^\star_{\text{r-nash}}, \pi')$. Then we can further write

$$
\begin{aligned}
p^\star(\pi_{\text{rp-nash}}, \pi) \geq& \min_{p \in \mathcal{P}} p(\pi^\star_{\text{r-nash}}, \widetilde{\pi}) - p^\star(\pi^\star_{\text{r-nash}}, \widetilde{\pi}) + 0.5 \\
\geq& 0.5 - \min_{p \in \mathcal{P}} \sqrt{C \mathbb{E}_{x \sim \mathcal{D}, y, y' \overset{\text{i.i.d.}}{\sim} \pi_{\text{data}}(\cdot|x)} (p(x, y, y') - p^\star(x, y, y'))^2},
\end{aligned}
$$

where the first inequality is due to $\pi^\star_{\text{r-nash}} \in \arg\min_{\pi' \in \Pi(\pi_{\text{data}}, C)} p^\star(\pi^\star_{\text{r-nash}}, \pi')$, and the second inequality follows from Equation 14. We can further upper bound this last term using $(p(x, y, y') - p^\star(x, y, y'))^2 = (\sqrt{p(x, y, y')} + \sqrt{p^\star(x, y, y')})^2 (\sqrt{p(x, y, y')} - \sqrt{p^\star(x, y, y')})^2 \leq 2^2 (\sqrt{p(x, y, y')} - \sqrt{p^\star(x, y, y')})^2$ as

$$
\begin{aligned}
p^\star(\pi_{\text{rp-nash}}, \pi) \geq& 0.5 - \min_{p \in \mathcal{P}} \sqrt{4 C \mathbb{E}_{x \sim \mathcal{D}, y, y' \overset{\text{i.i.d.}}{\sim} \pi_{\text{data}}(\cdot|x)} (\sqrt{p(x, y, y')} - \sqrt{p^\star(x, y, y')})^2} \\
\geq& 0.5 - 2 \min_{p \in \mathcal{P}} \sqrt{C \mathbb{E}_{x \sim \mathcal{D}} 2 H^2(p, p^\star)} \\
\geq& 0.5 - 2 \min_{p \in \mathcal{P}} \sqrt{C \mathbb{E}_{x \sim \mathcal{D}} 2 \text{TV}(p, p^\star)} \\
\geq& 0.5 - 2\sqrt{C\varepsilon},
\end{aligned}
$$

where the first inequality follows from the definition of Hellinger distance, second inequality from the relationship between Hellinger distance and total variation, and the last step is from our definition of $\mathcal{P}$. $\square$

## B.2 Bounded-likelihood-ratio-based coverage

Let $\Pi_{\sqrt{C}} = \{\pi \in \Pi \; : \; \|\pi/\pi_{\text{data}}\|_\infty \leq \sqrt{C}\}$. Then $\forall \pi, \pi' \in \Pi_{\sqrt{C}}$ and $\forall p_1, p_2 \in \mathcal{P}$

$$
\begin{aligned}
\mathbb{E}_{x \sim \mathcal{D}, y \sim \pi(\cdot|x), y' \sim \pi'(\cdot|x)} (p_1(x, y, y') - p_2(x, y, y'))^2 = \\
\mathbb{E}_{x \sim \mathcal{D}, y, y' \overset{\text{i.i.d.}}{\sim} \pi_{\text{data}}(\cdot|x)} \left[ \frac{\pi(y|x)\pi'(y|x)}{\pi_{\text{data}}(y|x)^2} (p_1(x, y, y') - p_2(x, y, y'))^2 \right] \\
\leq C \cdot \mathbb{E}_{x \sim \mathcal{D}, y, y' \overset{\text{i.i.d.}}{\sim} \pi_{\text{data}}(\cdot|x)} (p_1(x, y, y') - p_2(x, y, y')) x^2,
\end{aligned}
$$

which implies that $\Pi_{\sqrt{C}} \subseteq \Pi(\pi_{\text{data}}, C)$.

## B.3 Covariance-based coverage for linear preferences

Suppose $\mathcal{P} = \{w^T \phi(x, y, y') \; : \; w \in \mathcal{W}\}$ be a collection of linear preferences such that $p^\star \in \mathcal{P}$. Then the coverage condition (14) reduces to

$$
\mathbb{E}_{y \sim \pi(\cdot|x), y' \sim \pi'(\cdot|x)} ((w_1 - w_2)^\top \phi(x, y, y))^2 \leq C \mathbb{E}_{y, y' \overset{\text{i.i.d.}}{\sim} D_y(\cdot|x)} ((w_1 - w_2)^\top \phi(x, y, y))^2
$$

$$
\iff (w_1 - w_2)^\top \Sigma_{\pi, \pi'}(x)(w_1 - w_2) \leq C(w_1 - w_2)^\top \Sigma_{D_y, D_y}(w_1 - w_2),
$$

where we denote $\Sigma_{\pi, \pi'}(x) = \mathbb{E}_{y \sim \pi(\cdot|x), y' \sim \pi'(\cdot|x)} \phi(x, y, y) \phi(x, y, y)^\top$. This condition holds whenever we have

$$
\sup_u \frac{u^\top \Sigma_{\pi, \pi'} u}{u^\top \Sigma_{D_y, D_y} u} \leq C,
$$

which is an alignment condition between the covariances that is significantly weaker than the bounded density ratio condition necessitated by the definition of $\Pi_{\sqrt{C}}$.

## C Proof of Lemma 4.1

*Proof.* We consider the following objective for $\alpha \in [0,1]$ and a more general version, where we derive the objective for different values of KL regularization for the main and opponent policy, i.e., $\beta, \beta' \in \mathbb{R}^+$

$$\max_\pi \min_{p \in \mathcal{P}} \min_{\pi'} p(\pi, \pi') - \beta \mathrm{KL}\left(\pi \| \pi_{\mathrm{ref}}\right) + (1-\alpha)\beta' \mathrm{KL}\left(\pi' \| \pi_{\mathrm{ref}}\right) + \alpha \beta' \mathrm{KL}\left(\pi' \| \pi_{\mathrm{data}}\right)$$

Only looking at the inner minimization of $\pi'$, we get

$$\min_{\pi'} p(\pi, \pi') + (1-\alpha)\beta' \sum_y \pi'(y) \log \frac{\pi'(y)}{\pi_{\mathrm{ref}}(y)} + \alpha \beta' \sum_y \pi'(y) \log \frac{\pi'(y)}{\pi_{\mathrm{data}}(y)}$$

$$\min_{\pi'} p(\pi, \pi') + \beta' \sum_y \pi'(y) \log \frac{\pi'(y)}{\pi_{\mathrm{ref}}(y)^{1-\alpha} \pi_{\mathrm{data}}(y)^\alpha}$$

and thus, the optimal solution for $\pi'$ can be written as

$$\pi'_\star(y) = \frac{1}{Z} \pi_{\mathrm{ref}}(y)^{1-\alpha} \pi_{\mathrm{data}}(y)^\alpha \exp\left(-\frac{1}{\beta'} p(\pi, y)\right),$$

with partition function $Z = \sum_y \pi_{\mathrm{ref}}(y)^{1-\alpha} \pi_{\mathrm{data}}(y)^\alpha \exp\left(-\frac{1}{\beta'} p(\pi, y)\right)$. Plugging this back in the objective above gives

$$\max_\pi \min_{p \in \mathcal{P}} p(\pi, \pi'_\star) - \beta \mathrm{KL}\left(\pi \| \pi_{\mathrm{ref}}\right) + \beta' \sum_y \pi'_\star(y) \log \frac{\pi'_\star(y)}{\pi_{\mathrm{ref}}(y)^{1-\alpha} \pi_{\mathrm{data}}(y)^\alpha}$$

$$= \max_\pi \min_{p \in \mathcal{P}} p(\pi, \pi'_\star) - \beta \mathrm{KL}\left(\pi \| \pi_{\mathrm{ref}}\right) + \beta' \sum_y \pi'_\star(y) \left(-\frac{p(\pi, y)}{\beta'}\right) - \beta' \log Z$$

$$= \max_\pi \min_{p \in \mathcal{P}} -\beta \mathrm{KL}\left(\pi \| \pi_{\mathrm{ref}}\right) - \beta' \log Z$$

$$= \max_\pi \min_{p \in \mathcal{P}} -\beta \mathrm{KL}\left(\pi \| \pi_{\mathrm{ref}}\right) - \beta' \log \sum_y \pi_{\mathrm{ref}}(y)^{1-\alpha} \pi_{\mathrm{data}}(y)^\alpha \exp\left(-\frac{1}{\beta'} p(\pi, y)\right)$$

$$= \max_\pi \min_{p \in \mathcal{P}} -\beta \mathrm{KL}\left(\pi \| \pi_{\mathrm{ref}}\right) - \beta' \log \mathbb{E}_{y \sim \pi_{\mathrm{mix}}^\alpha} \exp\left(-\frac{1}{\beta'} p(\pi, y)\right) + \beta' \log Z',$$

where $\pi_{\mathrm{mix}}^\alpha(y) \propto \pi_{\mathrm{ref}}(y)^{1-\alpha} \pi_{\mathrm{data}}(y)^\alpha$ and $Z' = \sum_y \pi_{\mathrm{ref}}(y)^{1-\alpha} \pi_{\mathrm{data}}(y)^\alpha$ is a normalization constant, independent of optimization parameters. Dropping this term gives us an equivalent optimization objective in $\pi$. Setting $\beta' = \beta$ (which is usually the case) completes the proof of the lemma. $\square$

## D Proof of Lemma 4.2

*Proof.* Consider the log-sum-exp term with $\pi_{\mathrm{mix}}^\alpha(y) = \frac{1}{Z'} \pi_{\mathrm{ref}}(y)^{1-\alpha} \pi_{\mathrm{data}}(y)^\alpha$ and $Z' = \sum_y \pi_{\mathrm{ref}}(y)^{1-\alpha} \pi_{\mathrm{data}}(y)^\alpha$ as

$$= \log \mathbb{E}_{y \sim \pi_{\mathrm{mix}}^\alpha} \exp\left(-\frac{1}{\beta'} p(\pi, y)\right)$$

$$= \log \mathbb{E}_{y \sim \pi'} \left[\frac{\pi_{\mathrm{mix}}^\alpha(y)}{\pi'(y)} \exp\left(-\frac{1}{\beta'} p(\pi, y)\right)\right] \qquad (\pi' \text{ arbitrary})$$

$$\geq \mathbb{E}_{y \sim \pi'} \left[\log\left(\frac{\pi_{\mathrm{mix}}^\alpha(y)}{\pi'(y)} \exp\left(-\frac{1}{\beta'} p(\pi, y)\right)\right)\right] \qquad (\text{Jensen's inequality})$$

$$= -\frac{1}{\beta'} p(\pi, \pi') + \mathbb{E}_{y \sim \pi'} \log\left(\frac{\pi_{\mathrm{mix}}^\alpha(y)}{\pi'(y)}\right) = -\frac{1}{\beta'} p(\pi, \pi') - \mathrm{KL}\left(\pi' \| \pi_{\mathrm{mix}}^\alpha\right).$$

Setting $\beta' = \beta$ and taking the minimum over $p \in \mathcal{P}$ yields

$$\min_{p \in \mathcal{P}} -\log \mathbb{E}_{y \sim \pi_{\mathrm{mix}}^{\alpha}(\pi_{\mathrm{ref}}, \pi_{\mathrm{data}})} \left[ \exp\left(\frac{-p(\pi, y)}{\beta}\right) \right] \leq \min_{p \in \mathcal{P}} \frac{p(\pi, \pi')}{\beta} - \mathrm{KL}\left(\pi' \| \pi_{\mathrm{mix}}^{\alpha}(\pi_{\mathrm{ref}}, \pi_{\mathrm{data}})\right). \tag{15}$$

Choosing $\pi' = \pi_{\mathrm{mix}}^{\alpha}(\bar{\pi}, \pi_{\mathrm{data}})$ gives the desired result with

$$\kappa = -\mathrm{KL}\left(\pi_{\mathrm{mix}}^{\alpha}(\bar{\pi}, \pi_{\mathrm{data}}) \| \pi_{\mathrm{mix}}^{\alpha}(\pi_{\mathrm{ref}}, \pi_{\mathrm{data}})\right).$$

$\square$

# E   Pessimistic Reward-based Policy Optimization

A straightforward way to simplify our general preference-based algorithm P3O is to replace the general preference function with a BTL reparameterization, $p_{\mathrm{BTL}}(r)$, that uses an underlying reward function $r$. This substitution yields the objective in (13). We provide modified pseudo-code in Algorithm 2, where we also use the preference learning rate $\eta_p$ as the reward function's learning rate.

Building on Azar et al. (2023), we can further consider a monotonically increasing function $\Psi : [0, 1] \to \mathbb{R}$, leading to the modified objective:

$$\max_{\pi} \min_{r} \mathbb{E}_{y \sim \pi, \, y' \sim \pi_{\mathrm{mix}}^{\alpha}(\bar{\pi}, \pi_{\mathrm{data}})} \left[ \Psi\left(p_{\mathrm{BTL}}(y, \, y'; \, r)\right) \right]$$
$$- \beta \mathrm{KL}\left(\pi \| \pi_{\mathrm{ref}}\right) - \lambda \mathrm{KL}_{\pi_{\mathrm{data}}}\left(p_{\mathrm{BTL}}(r_{\mathrm{mle}}) \, \| \, p_{\mathrm{BTL}}(r)\right).$$

When $\Psi$ is the identity function, we recover PRPO. By contrast, setting $\Psi(q) = \ln\left(q/(1-q)\right)$ produces the objective

$$\max_{\pi} \min_{r} \mathbb{E}_{y \sim \pi, \, y' \sim \pi_{\mathrm{mix}}^{\alpha}(\bar{\pi}, \pi_{\mathrm{data}})} [r(y) - r(y')]$$
$$- \beta \mathrm{KL}\left(\pi \| \pi_{\mathrm{ref}}\right) - \lambda \mathrm{KL}_{\pi_{\mathrm{data}}}\left(p_{\mathrm{BTL}}(r_{\mathrm{mle}}) \, \| \, p_{\mathrm{BTL}}(r)\right).$$

This latter form matches existing pessimistic reward-based methods (Fisch et al., 2024; Liu et al., 2024), although those works often fix the opponent (rather than using $\pi_{\mathrm{mix}}^{\alpha}$) and employ a log-likelihood term to maintain a version space of plausible reward functions. Both Liu et al. (2024) and Fisch et al. (2024) circumvent the inner minimization by solving it in closed form.

**Geometric mixture** $\pi_{\mathrm{mix}}^{\alpha} \propto \bar{\pi}_t^{1-\alpha} \pi_{\mathrm{data}}^{\alpha}$. In Algorithms 1 and 2, the geometric mixture of policy is implemented as logit mixing, i.e., for the token at step $i$, i.e., $y_i$, let $\overrightarrow{h}_{i,t}$ be the vector of logits generated for sampling a token at the $i'$th step by model for $\pi_t$, similarly $\overrightarrow{h}_{i,\mathrm{data}}$ be the vector of logits generated for sampling a token at the $i'$th step by model for $\pi_{\mathrm{data}}$. The final set of vector of logits then used to sample the token at $i'$th step is $\overrightarrow{h}_{i,\mathrm{mix}}^{\alpha} = (1-\alpha)\overrightarrow{h}_{i,t} + \alpha\overrightarrow{h}_{i,\mathrm{data}}$. Hence, the token at step $i$ is sampled from the distribution induced by logits $\overrightarrow{h}_{i,\mathrm{mix}}^{\alpha}$ representing the mixture of the two LLMs.

# F   Tabular Experiments

**Tabular experiments summary.** We first conduct experiments in a controlled, tabular setting with three possible outputs $y \in \{y_1, y_2, y_3\}$, and a ground-truth preference matrix $p^{\star}$. We vary the probability of sampling $y_3$ from 0.0 to 0.2, distributing the remaining probability equally among $(y_1, y_2)$. Thus, $y_3$ is consistently the under-sampled output. We conduct experiments with varying Nash strategies, including cases where $y_3$ is preferred, as well as dispreferred over the other two actions.

Figure 4 illustrates a case in which the under-sampled action is dispreferred ($\pi_{\mathrm{nash}}(y_3) = 0.1$). We compare Exact P3O (0.1) (EP3O (0.1)), EP3O (0), as defined in Eq. (10), and the non-pessimistic Nash policy obtained from $p_{\mathrm{mle}}$. Additional scenarios, including ones where the under-sampled action is genuinely preferred,

---

**Algorithm 2** Pessimistic Reward-based Policy Optimization (PRPO ($\alpha$))

---

**Hyperparameters:** Mixing coefficient $\alpha$, policy regularization coefficient $\beta$, preference regularization coefficient $\lambda$, exponential moving average parameter $\gamma$, learning rates $(\eta_p, \eta_\pi)$

**Initialize:** $\bar{\pi}_1 = \pi_1 = \pi_{\text{ref}}$, $r_1 = r_{\text{mle}}$

**for** $t = 1, 2, \ldots$ **do**

    Set $\pi_{\text{mix}}^\alpha \propto \bar{\pi}_t^{1-\alpha} \pi_{\text{data}}^\alpha$ as mix of $\pi_{\text{ref}}$ and EMA $\bar{\pi}_t$ for restricted Nash,

    Approximate current objective (13):

$$J_{\text{PRPO}(\alpha)}(\pi_t, r_t) \doteq p_{\text{BTL}}\left(\pi_t, \pi_{\text{mix}}^\alpha; r_t\right) - \beta \text{KL}\left(\pi_t \| \pi_{\text{ref}}\right) - \lambda \text{KL}_{\pi_{\text{data}}}\left(p_{\text{BTL}}(r_{\text{mle}}) \| p_{\text{BTL}}(r)\right)$$

    Update $\pi_{t+1} \leftarrow \pi_t + \eta_\pi \left. \frac{\partial J_{\text{PRPO}(\alpha)}(\pi, r_t)}{\partial \pi} \right|_{\pi = \pi_t}$

    Update $r_{t+1} \leftarrow r_t - \eta_p \left. \frac{\partial J_{\text{PRPO}(\alpha)}(\pi_t, r)}{\partial r} \right|_{r = r_t}$

    Update $\bar{\pi}_{t+1} \leftarrow \gamma \pi_t + (1 - \gamma) \bar{\pi}_t$

**end for**

---

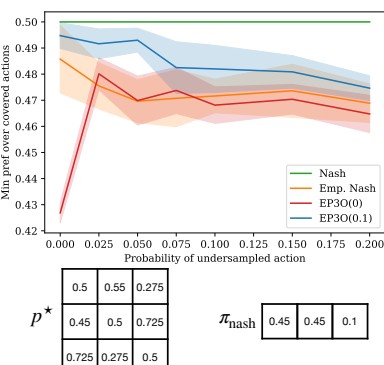

Figure 4: **Tabular experiments:** Comparison of the different objectives with an explicit search over the main policy, opponent policy, and version space. The $X$-axis shows the probability assigned to the under-sampled output ($y_3$), from $0.0 \to 0.2$. The $Y$-axis indicates the minimum preference of the policy found over all covered actions (higher is better). We also show the ground-truth preference matrix $p^\star$ (bottom-right) and the corresponding $\pi_{\text{nash}}$ (bottom-left). Results are averaged over 10 random seeds, shaded areas represent $\pm 2\times$ std error. EP3O(0.1) corresponding to the restricted Nash formulation consistently does well, particularly when the sampling rate of $y_3$ is very low (left part of the plot).

appear in Figure 5. Notably, EP3O (0.1) serves as a robust default choice: when the under-sampled action is truly dispreferred (particularly under extremely low sampling), the restricted pessimism in EP3O (0.1) prevents the policy from overcommitting to an insufficiently explored but suboptimal action. Conversely, if the under-sampled action is actually favored under the true $\pi_{\text{nash}}$, even a small amount of data may guide non-pessimistic methods to weight that action correctly—potentially yielding strong performance. Since we typically lack ground-truth preferences, EP3O (0.1) offers a "safe-default" strategy.

**Detailed description.** To illustrate our approach and evaluate the proposed objectives, we conduct experiments in a tabular setting with three possible outputs $y \in \{y_1, y_2, y_3\}$, and a ground-truth preference matrix $p^\star$. We vary the probability of sampling $y_3$ from $0.0 \to 0.2$, distributing the remaining probability equally among $(y_1, y_2)$. Thus, $y_3$ is consistently the under-sampled output. Under each sampling policy, we collect 500 action pairs and use $p^\star$ to sample their pairwise preferences, forming a preference dataset. From this dataset, we estimate the empirical preference model $p_{\text{mle}}$. We then define an uncertainty set $\mathcal{P}(p_{\text{mle}}, c)$[6] $\subset \{\mathcal{Y} \times \mathcal{Y} \to [0, 1]\}$ around $p_{\text{mle}}$ by enumerating all preferences satisfying:

$$\mathcal{P}(p_{\text{mle}}, c) = \left\{ p : p_{\text{mle}}^{c-}(y, y') \leq p(y, y') \leq p_{\text{mle}}^{c+}(y, y') \right\},$$

$\forall y \neq y'$ where,

$$p_{\text{mle}}^{c-}(y, y') := \max\left(p_{\text{mle}}(y, y') - c\,\sigma(y, y'), 0\right),$$
$$p_{\text{mle}}^{c+}(y, y') := \min\left(p_{\text{mle}}(y, y') + c\,\sigma(y, y'), 1\right),$$

---

[6]Note that this differs from our earlier definition in (7), which took the preference dataset $\mathcal{D}$ as the first argument.

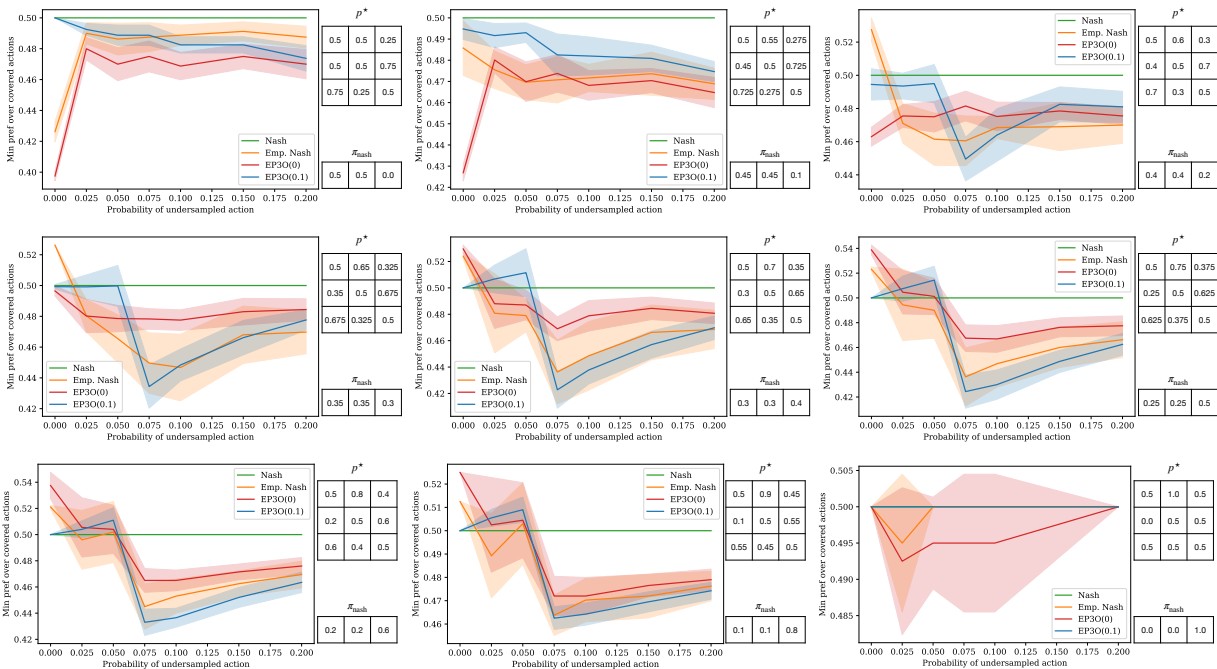

Figure 5: **Tabular experiments** (Continued from Figure 4) Comparison of the different objective functions with an explicit search over the main policy, opponent policy, and version space. The $X$-axis shows the probability assigned to the under-sampled output ($y_3$), from $0.0 \to 0.2$. The $Y$-axis indicates the minimum preference of the policy found over all covered actions (higher is better). Each plot corresponds to a differernt $p^\star$ setting. Each figure also shows the ground-truth preference matrix $p^\star$ (on the top-right) and the corresponding $\pi_{\mathrm{nash}}$ (bottom-right) to each plot. Results are averaged over 10 random seeds, shaded areas represent $\pm 2\times$ standard error. EP3O(0.1) corresponding to the restricted Nash formulation consistently does well, particularly when the undersampled output is dispreferred and its sampling rate is very low (left part of plot).

and $\sigma(y, y')$ is the empirical standard deviation. Note that, $\mathcal{P}(p_{\mathrm{mle}}, 0)$ is a singleton set $\{p_{\mathrm{mle}}\}$. We then optimize the objective in (10) (for $\alpha = 0.1$ and $\alpha = 0$) via a brute-force search over the main policy, the opponent policy, and the version space of preference models. We specify $\pi_{\mathrm{ref}}$ to the uniform random policy. In each case, we choose the smallest $c^\star$ such that $p^\star$ lies in $\mathcal{P}(p_{\mathrm{mle}}, c^\star)$:

$$c^\star = \arg\min_c \{c \mid p^\star \in \mathcal{P}(p_{\mathrm{mle}}, c)\}.$$

**Brute-force optimization:** We perform a grid search over the main policy, the opponent policy, and all possible preference matrices in the version space. Each policy is discretized into 11 points per action, resulting in $11^3 = 1331$ possible policies for each player. Similarly, we discretize each entry of the preference matrix between $p_{\mathrm{mle}}^{c-}$ and $p_{\mathrm{mle}}^{c+}$ into 11 points. Because the matrix is fully specified by three parameters, this again yields $11^3 = 1331$ possible matrices to search over. To calculate the minimum preference over preferred actions, we drop any action with a probability below 0.05 for the restricted action set.

# G   Additional Results

## G.1   Pairwise win-rates with confidence intervals

A copy of our experimental evaluation in Table 1 with 95% bootstrap CIs is included in Table 2.

| Method | Summarization Win-rate ↗ vs. | | | Helpfulness Win-rate ↗ vs. | | |
|---|---|---|---|---|---|---|
| | $\pi_{\text{ref}}$ | P3O | PRPO | $\pi_{\text{ref}}$ | P3O | PRPO |
| P3O | 0.91 (0.89, 0.92) | 0.50 (0.50, 0.50) | 0.57 (0.55, 0.59) | 0.89 (0.87, 0.91) | 0.50 (0.50, 0.50) | 0.59 (0.56, 0.62) |
| PRPO | 0.88 (0.86, 0.89) | 0.43 (0.41, 0.45) | 0.50 (0.50, 0.50) | 0.86 (0.83, 0.88) | 0.41 (0.38, 0.43) | 0.50 (0.50, 0.50) |
| NashEMA | 0.83 (0.81, 0.84) | 0.31 (0.29, 0.33) | 0.40 (0.37, 0.42) | 0.90 (0.87, 0.92) | 0.67 (0.64, 0.70) | 0.76 (0.73, 0.79) |
| IterDPO | 0.89 (0.88, 0.90) | 0.44 (0.41, 0.46) | 0.52 (0.50, 0.55) | - | - | - |
| REINFORCE | 0.84 (0.82, 0.85) | 0.35 (0.34, 0.37) | 0.43 (0.41, 0.45) | 0.78 (0.76, 0.81) | 0.46 (0.43, 0.48) | 0.54 (0.51, 0.56) |
| DPO | 0.82 (0.80, 0.84) | 0.34 (0.32, 0.36) | 0.41 (0.40, 0.43) | 0.73 (0.70, 0.76) | 0.31 (0.29, 0.34) | 0.38 (0.35, 0.41) |
| WARM | 0.74 (0.72, 0.76) | 0.26 (0.24, 0.28) | 0.31 (0.29, 0.33) | - | - | - |
| PDPO | 0.83 (0.81, 0.85) | 0.35 (0.33, 0.37) | 0.42 (0.40, 0.44) | 0.71 (0.68, 0.74) | 0.25 (0.22, 0.28) | 0.31 (0.29, 0.34) |

Table 2: Win-rates with 95% confidence intervals over the evaluation dataset.

### G.2 Training vs. evaluation dynamics with and without pessimism

Figure 6 presents learning curves on the summarization and helpfulness tasks for different methods over a period of 20,000 training steps. As we note in the curves, the far left side corresponds to the starting point where $\pi_1 \doteq \pi_{\text{ref}}$, and hence the initial preference is 0.5. In the left figure we can see that (nearly) all methods consistently seem to be improving on the training reward, where in fact REINFORCE actually seems to be doing *better* than pessimistic methods. However, that ordering is not followed when evaluated with the much bigger evaluation preference model (i.e., Gemini 1.5), as seen in the right figure, whereas our pessimistic methods (P3O, PRPO) outperform the standard RLHF methods, and do not degrade over time.

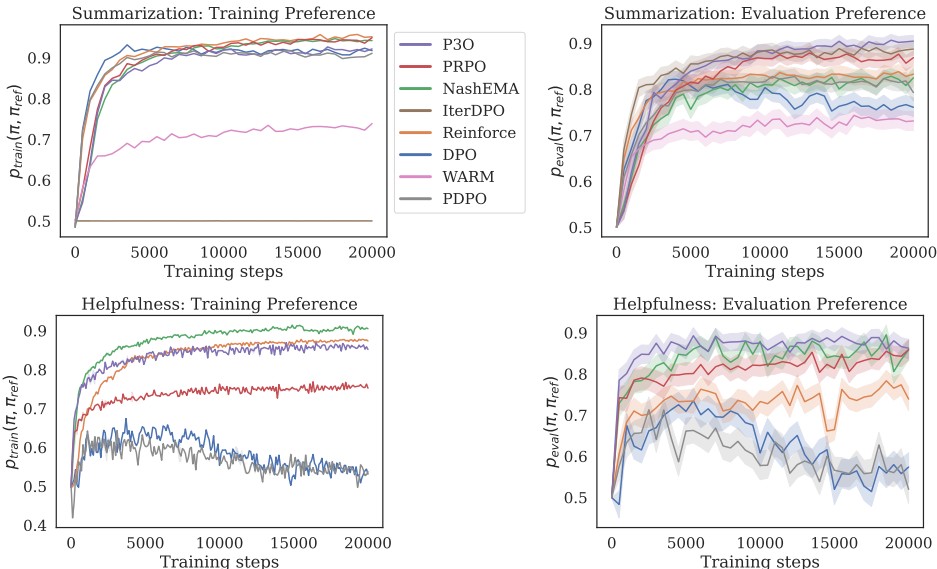

Figure 6: **Training vs. evaluation dynamics for summarization and helpfulness.** The horizontal axis shows the number of training steps, and the vertical axis shows the preference of the learned policies against $\pi_{\text{ref}}$. We use the same hyperparameters for each method as in Figure 1. (**Left**) The training preference is measured by $p_{\text{mle}}$ for preference-based methods and by $p_{\text{BTL}}(r_{\text{mle}})$ for RLHF methods. (**Right**) The evaluation preference measured is by Gemini 1.5 Flash (repeated from Figure 1).

### G.3 Parameter efficient training

As pointed out in the main text, the resulting pessimistic algorithm can have a considerable amount of overhead in terms of memory in light of it requiring to maintain an additional copy of the policy model as well as a reward model with optimizer state to perform the pessimistic update. In this experiment we freeze all the parameters of the reward model and train only the attention heads (specifically the encoder-decoder attention parameters in the T5 architecture), this results in training 14% of the total model parameters

(note that the reward models tend to be the bigger models in this setup, as otherwise they can be easily hacked by a bigger policy model). We refer to this variant of P3O as P3O PET. Figure 7 presents results for this scenario where P3O PET retains full performance that can be observed with the P3O while being much more memory friendly, hence providing us with a much more practical variant of the algorithm.

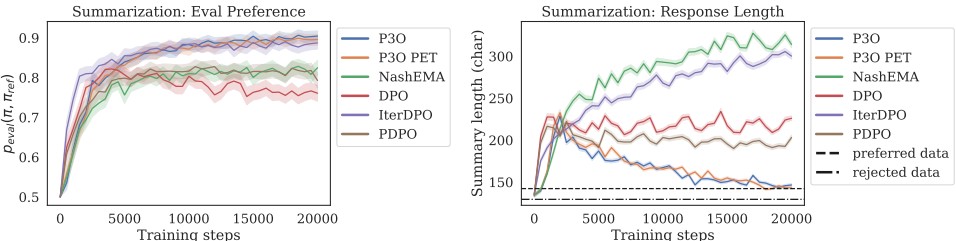

Figure 7: **Results on parameter efficient training (PET).** (**Left**) Gemini preference win-rates for P3O and P3O PET. (**Right**) Summary length produced by each method. P3O PET retains full performance that can be observed by P3O while being much more memory efficient.

### G.4 Non-pessimistic methods often require heavy KL regularization to avoid reward hacking

In Figure 8 we provide example ablations of the KL regularization parameter $\beta$ for both the REINFORCE and DPO baselines on the summarization task. Unless high values of $\beta$ are used, the policy becomes hacked and degrades. For REINFORCE in particular, where a reward model is used during training, we can further see that the *training* preference based on the reward model continues to increase throughout training, even when the evaluation preferences are rapidly declining, see Figure 9. Similar results can also be observed for the other non-pessimistic RLHF methods, where higher regularization is often required for best performance: see Appendix G.7 for a detailed description of all the hyperparameters we tested and their best values.

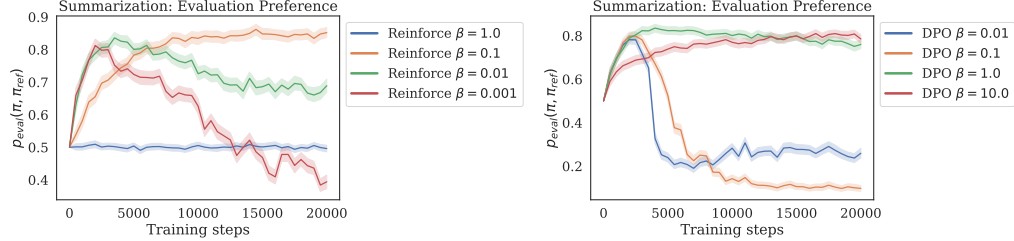

Figure 8: **Illustrative ablation of KL regularization for RLHF methods on summarization.** The horizontal axis indicates the number of training steps, and the vertical axis shows the preference of the learned policies against $\pi_{\mathrm{ref}}$, as measured by Gemini 1.5 Flash.

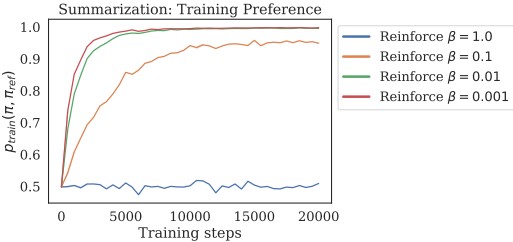

Figure 9: **Reward hacking in the case of REINFORCE on summarization.** The horizontal axis shows the number of training steps, and the vertical axis shows the preference of the learned policies against $\pi_{\text{ref}}$ using $p_{\text{BTL}}(r_{\text{mle}})$. Comparing with evaluation preferences in Figure 8 (left), REINFORCE exhibits reward hacking particularly for $\beta \in \{0.01, 0.001\}$, where the preference on the training reward seems to be stable, i.e., close to 1, but the evaluation preference is degrading.

### G.5    Evolution of the KL divergence over training

Pessimistic methods consistently demonstrate lower KL divergence (together with strong evaluation win-rates, per Figure 1) compared to other approaches, despite utilizing significantly smaller values of $\beta$. An interesting exception, however, is REINFORCE on the summarization task: There REINFORCE maintains a lower KL divergence to $\pi_{\text{ref}}$ than the pessimistic methods. On the other hand, P3O and PRPO move farther in distribution space than REINFORCE, but clearly along dimensions that improve perceived quality. Note that while WARM also has a very low KL divergence, it is over-regularized, and does not perform well according to evaluation preference.

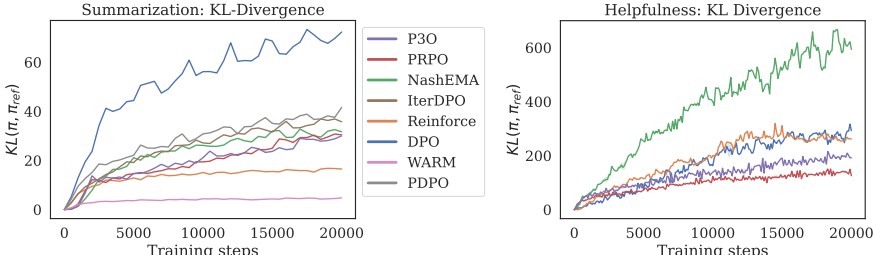

Figure 10: **KL divergence over training.** The horizontal axis represents the number of training steps, while the vertical axis indicates the KL divergence of the learned policy from $\pi_{\text{ref}}$.

### G.6 Robustness to evaluation preference on the helpfulness task

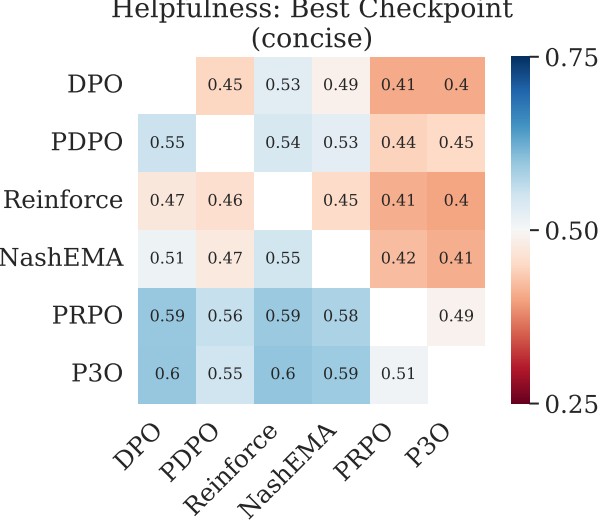

Figure 11: **Evaluation preference robustness on helpfulness.** Pairwise evaluations across the different methods when using an evaluation that emphasizes both helpful *and* concise prompts for the helpfulness tasks. See the "Helpfulness Concise Evaluation Prompt" shown in Section G.7. This results in much worse performance for the length-hacked REINFORCE and NashEMA models, whereas the non-length-hacked pessimistic methods P3O and PRPO still perform very well—demonstrating the better robustness that training with pessimism affords to train/eval preference mismatches.

### G.7 Detailed setup of the empirical evaluation

**Hyper-parameters**: All policies are trained for 20,000 steps, where each steps corresponds to a gradient step performed on a given mini-batch. Tables 3-6 present the hyperparameters swept over for different methods. Parameters in bold (over sweeps) were the final ones used.

Table 3: Hyperparameters for P3O  and PRPO  for Summarization (TL;DR)

| Hyperparameter | Value / Range |
|---|---|
| Training Steps | 20,000 |
| Mini-Batch Size | 32 |
| Policy Learning Rate $\eta_\pi$ | $10^{-5}$ |
| Preference Learning Rate $\eta_p$ | $\{\mathbf{2.5 \times 10^{-5}},\ 5 \times 10^{-5}\}$ |
| Regularization Coefficient $\beta$ | $10^{-5}$ |
| $\lambda$ (Sweep) | $\{1,\ 2,\ 4,\ 8,\ 16,\ \mathbf{32},\ 64\}$ |
| EMA Parameter $\gamma$ | 0.0025 |
| Policy Mixing $\alpha$ | $\{\mathbf{0.0}, 0.01, 0.1, 0.25, 0.5\}$ |
| Context Length | 1024 |
| Generation Length | 128 |

Additionally, for WARM, IterDPO, and PDPO we swept the following hyperparameters:

- Regularization coefficient sweep for Reinforce with WARM: $\{10^{-4}, 10^{-3}, 10^{-2}, \mathbf{0.1}, 1.0, 10.0\}$.

- Regularization coefficient sweep for IterDPO: $\{10^{-4}, 10^{-3}, 10^{-2}, 0.1, 1.0, \mathbf{10.0}\}$.

Table 4: Hyperparameters for RLHF methods (DPO, REINFORCE, NashEMA) for Summarization.

| Hyperparameter | Value / Range |
| --- | --- |
| Training Steps | 20,000 |
| Mini-Batch Size | 32 |
| Policy Learning Rate $\eta_\pi$ | $10^{-5}$ |
| Regularization Coefficient $\beta$ (Sweep) | $\{10^{-5}, 10^{-4}, 10^{-3}, \mathbf{10^{-2}}, \mathbf{0.1}, \mathbf{1.0}, 10.0\}$ |
| Context Length | 1024 |
| Generation Length | 128 |

- Regularization coefficient sweep for PDPO: $\{10^{-2}, 10^{-1}, 1.0, \mathbf{3.0}, 10.0, 30.0, 100.0\}$.

Table 5: Hyperparameters for P3O and PRPO for Helpfulness.

| Hyperparameter | Value / Range |
| --- | --- |
| Training Steps | 20,000 |
| Mini-Batch Size | 16 |
| Policy Learning Rate $\eta_\pi$ | $10^{-5}$ |
| Preference Learning Rate $\eta_p$ | $5 \times 10^{-5}$ |
| Regularization Coefficient $\beta$ | $10^{-5}$ |
| $\lambda$ (Sweep) | $\{32, \mathbf{64}\}$ |
| EMA Parameter $\gamma$ | 0.0025 |
| Policy Mixing $\alpha$ | $\{\mathbf{0.0}, 0.5\}$ |
| Context Length | 1024 |
| Generation Length | 128 |

Table 6: Hyperparameters for RLHF methods (DPO, REINFORCE, NashEMA) for Helpfulness

| Hyperparameter | Value / Range |
| --- | --- |
| Training Steps | 20,000 |
| Mini-Batch Size | 16 |
| Policy Learning Rate $\eta_\pi$ | $10^{-5}$ |
| Regularization Coefficient $\beta$ (Sweep) | $\{\mathbf{10^{-4}}, \mathbf{10^{-3}}, 10^{-2}, 0.1, 1.0, \mathbf{10.0}\}$ |
| Context Length | 1024 |
| Generation Length | 128 |

**Evaluation:** We save a checkpoint for policies at every 500 steps, and generate summaries from $\pi_t$ for evaluation. To evaluate the learned model, we query Gemini 1.5 Flash (Gemini Team, 2024) to judge which summary is better for the given input context. The evaluation prompts are shown below.

> **Summarization Evaluation Prompt:** You are an expert summary rater who prefers very short and high quality summaries. Given a document and two candidate summaries, say 1 if SUMMARY1 is the better summary, or 2 if SUMMARY2 is the better summary. If neither one is better than the other, say 0. Give a short reasoning for your answer.
> ARTICLE: <article-here >
> SUMMARY1: <summary-by-$\pi_t$>
> SUMMARY2: <summary-by-$\pi_{\text{ref}}$>.

**Helpfulness Evaluation Prompt:** You are an expert rater of AI Assistant responses. Given a dialogue history and two candidate AI Assistant responses, say 1 if RESPONSE1 is more helpful, or 2 if RESPONSE2 is more helpful. If neither one is more helpful than the other, say 0. Give a short reasoning for your answer.
Dialogue: <context-here >
RESPONSE1: <response-by-$\pi_t$>
RESPONSE2: <response-by-$\pi_{\text{ref}}$>.

**Helpfulness Concise Evaluation Prompt:** You are an expert rater of AI Assistant responses who prefers responses which are helpful for a human reader, but not overly verbose. Given a dialogue history and two candidate AI Assistant responses, say 1 if RESPONSE1 is a better helpful response that is not overly verbose, or 2 if RESPONSE2 is a better helpful response that is not overly verbose. If neither one is better than the other, say 0. Give a short reasoning for your answer.
Dialogue: <context-here >
RESPONSE1: <response-by-$\pi_t$>
RESPONSE2: <response-by-$\pi_{\text{ref}}$>.

To avoid positional bias, we make two queries for each comparison with swapped response orders.

### G.8 Computation requirements

Experiments were run on the TPUv6e platform, where each single run for the smaller summarization models took less than a day for 20,000 steps. The total compute can be multiplied by the sweep parameters, which would be close to 150-180 TPUv6e days, which also accounts for failed runs. For the larger helpfulness models, training was approximately $3\times$ slower.

## H  Additional Discussion

**Choice of base models.**

The choice of T5 and PaLM-based architectures for the policy and reward models follows the experimental setup of established studies on reward hacking for these datasets (Eisenstein et al., 2024; Fisch et al., 2024). This allows us to cleanly isolate the effect of our algorithmic contribution (pessimism) from architectural confounders, ensuring a fair comparison of objective functions.

**Output diversity and mode collapse.**  A natural concern with adversarial max-min objectives is the risk of mode collapse, reminiscent of GAN-style training. In our formulation, however, the max-player $\pi$ is KL-regularized toward the reference policy $\pi_{\text{ref}}$, which discourages entropy collapse—making P3O no more susceptible to this issue than any standard RLHF method. In fact, the empirical evidence suggests the opposite: standard methods often collapse into narrow, repetitive stylistic hacks (e.g., length inflation or exhaustive lists), whereas P3O and PRPO maintain more natural and diverse generation patterns. Qualitative examples in Appendix I further corroborate this observation.

## I  Sample Generations

The following are a number of sample generations from the best checkpoints with each method on both the summarization and helpfulness datasets (distinguished as "Post" vs. "Dialogue" for summarization vs. helpfulness, respectively). In agreement with Figure 3, DPO, REINFORCE and NashEMA tend to give longer and more extractive responses on the summarization task, while the coherency of DPO's summaries also suffers. Both PRPO and P3O tend to give shorter, succinct summaries. Meanwhile, on the helpfulness task, broadly speaking, one can see from the generations that the selected DPO model (with high $\beta$ regularization) does not deviate too far from the reference model, while the REINFORCE and NashEMA models nearly

always gives long, wordy responses that try to be overly informative, with the REINFORCE model in particular also exhibiting a high prevalence of repeated key-words, phrasings, or tokens ("You are right", "sure", "*", lists). PRPO and P3O generations tend to be more detailed than the reference (though it is not immune to hallucination), and use lists only when appropriate.

---

**Post 1**

I'll try to keep this short!

**Background**

- I've always been an on again/off again (very casual!) jogger, typically doing 3 - 5 k
- My knees have always been finicky, and I went to a physio who thought I had "runner's knee"
- Pre-pregnancy, my "runner's knee" would flare up when I got to the 8 - 10 k distance range, even if I had a decent base (doing a C210k type program)

**Current Problem**

I had my baby a year ago, so all in all I haven't run for about 1.5 years. I'm quite slim and have been doing aerobics-style classes for the past year, so I'm not totally out of shape. Body weight exercises, aerobics, bikes and ellipticals are all fine. However, when I run even the tinyiest bit, or even go on a long walk or a hike, my pelvis gets very sore and tight, and my knees start hurting very quickly. I already am doing general squats/lunges/stretching type things.

I'm starting to feel like running just isn't for me anymore. Which is a bummer, because I really enjoy running!

Has anyone had something similar? Can anyone recommend some stretches or exercises that might help? Should I see a Dr? Or should I just see a physio? Not quite sure how to proceed.

Thanks!

---

**DPO:** nephew-jogger hasn't run for 1.5 years, knees become very sore and tight very quickly after running, making running no longer enjoyable. Has anyone gone through this before and have any advice?

---

**REINFORCE:** I've been casually jogging for 1.5 years and recently my pelvis and knees are really sore after running. Body weight exercises - aerobics work (all good!) - don't really enjoy running anymore. Any advice/stretches that might help?

---

**NashEMA:** Don't run much anymore after having a baby, and when I try to run even a tiny bit it causes my pelvis to get very sore and my knees get very sore and painful. How can I fix this? Ideally, I'd like some stretches and exercises that might help. Does anyone have any advice? Or should I just see a physio?

---

**PRPO:** Don't run much anymore, pelvis gets very sore and knees hurt very quickly when I do. Need help/advice!

---

**P3O:** Don't run much recently, pelvis and knees get very sore and very quickly when running; any tips or exercises that might help?

**Post 2**

So here's my situation. I love www.omegle.com, it's a great site to have some good conversations and meet some awesome people. About a week ago, I was on said site, and I met this girl. Named Sammie. We had a really good conversation. And I asked her to email me, she responded by telling me to add her on facebook. I though (what have I got to lose?)

So I did. Holy Gorgeous Reddit.

Well I've talked to her quite a lot, and I actually really like her. However I have a nagging suspicion that all is not as she says it is... She has 130 someodd friends, the majority of which have <9 friends!! She doesn't have a single friend from her hometown. All of her friend's posts on her wall are exactly the same. They use the same format, all have little quirks that hint to me that they are all the same person!!!

Now this is all well and good, I could simply ask her for a phone number, or an address, and confirm my suspicions. But I'm better than that =) Luckily I'm a programmer.

So here's what I'm thinking. Simple detective program to discover her identity.

If she isn't who she says she is, I'd still like to talk to her, I just don't want to be lied to, you know?

**DPO:** Girl I met on Omgle.com introduced herself on facebook, have suspicions that she has 130 someodd facebook friends which all match her exact profile details. Need simple detective program.

**REINFORCE:** I'm curious as to how to uncover a beautiful girl's identity via facebook. All the friends on her wall are completely identical and hint at her possibly being the same person. Is there any simple program I can write to discover her identity?

**NashEMA:** I met this girl on omegle, like her, but I have a suspicion that all of her Facebook friends are the same person. All of the posts on her wall are extremely similar and tell me that she may be a scammer. Potential detective program to find her identity. Please help!

**PRPO:** I like this girl on omegle, like her, have suspicion she may not be who she says she is... How can I create simple detective tool to confirm her identity?

**P3O:** I like this girl on omegle, but her facebook friends strongly hint that she isn't who she says she is. Simple detective program to discover her identity?

**Post 3**

Hi /r/Relationships,

I'm a 25 year old male and I've never been in a relationship and I'm starting to wonder if I ever will.

I have a decent job, a car, a home. I'm successful in every part of my life apart from the part that seem to matter the most to other people. I've never been in love and I've never been in a relationship with a woman.

It's getting harder and harder as time goes on too. When I was 23 I went on a date and it ended up with the girl in my bed, but I chickened out and just went to sleep. It's stupid to be a virgin at 22, and I didn't want to look stupid nor admit that fact. Now I'm 25 and I just feel even more ridiculous. I mean, the pool of people who haven't done any of this kind of stuff by my age must be pretty damn small. Most people won't encounter them.

Sometimes I think about trying online dating, but in the end I'm going to come up against that exact same wall that I did 2 years ago.

Then there's part of me that wonders if I can be bothered with it all. I look around at all my friends and there seems to be a huge amount of interdependence and compromise in relationships. I've always been really independent and liked my own space, so I'm not sure how well I'd even function in a relationship.

My family all seem to be convinced I'm gay, which is frustrating. My reluctance to talk about it with them is basically proof. I don't even know what's worse at this point.. Coming out as gay would probably be easier and more normal at this point.

I feel like I'm missing a huge part of the life experience and that I'm emotionally stunted compared to my peers. It's not like I'm even an introvert, I'm very extroverted according to all my friends.

So yeah, I don't know what to do. Has anyone been in a similar situation of felt the same way? How did you get over it? Did you get over it?

Thanks in advance.

**DPO:** Met a virgin 2 years ago when I was 23, dating has become harder and harder as time passes and I'm starting to wonder if I ever will be in a relationship. Is coming out as gay a good idea? How did you get over it?

**REINFORCE:** I'm 25 and I've never been in a relationship. Like my peers all seem to think I'm gay. I essentially feel awful about my overall life experience. I've never been in a relationship. Advice on what I should do? Thanks for your input!

**NashEMA:** I'm a 25 year old male who has never been in a relationship, and I feel really depressed about it. I'm not sure how well I'd be able to function in a relationship. My family thinks I'm gay. Not sure how to handle myself or if I should even get involved in a relationship. Looking for help on how to fix this.

**PRPO:** I've never been in a relationship and I'm starting to wonder if I'll ever get over it. How did you deal?

**P3O:** I've never been in a relationship and I'm starting to wonder whether I will ever be in one. How did you get over it?

**Post 4**

I've been taking a tax course in university and I was thinking of ways to reduce my tax liability for the year. I had a good paid internship for most of the year so my income is fairly high and even though basic calculations have me receiving a refund for this tax year overall, it can still be increased.

My question is on common law relationships. Under my federal and provincial tax forms non-refundable credits can be claimed for a spouse or common law partner. For 2013 this will amount to 11,038 - the Net Income of said person. Technically I do not have a common law partner, just a long-term girlfriend who does not live with me (she lives at home and has very little income for 2013). If I were to claim her as a common law partner it would boost my refund by over 25% by my calculations. Would doing this be viable or is this something that can actually be audited by the CRA, what additional impact may this have on my return?

If you have any additional tax avoidance advice etc. for Canada that is also welcome! Especially thoughts that are not fairly obvious (like RRSP contributions).

**DPO:** Bought a fairly high tax year internship, income fairly high, common law partner non-refundable credit ($11038) is available for non-living girlfriend, could I claim her as common law partner for tax rebate boost?

**REINFORCE:** I have a pretty high income and have a good internship, for tax year 2013 the non-refundable credit for a common law partner will potentially boost my refund by over 25%. Also, any additional tax insight is welcome!

**NashEMA:** I have quite high income for this tax year, would claiming a common law partner to boost my refund by 25% if I did so. I do not have a common law partner, but I would like to know if it is viable in terms of tax avoidance for Canada. Would it be viable and if so what impact would this have?

**PRPO:** My federal tax forms allow for common law partner credits to be claimed. Would claiming my long-term girlfriend as such boost my refund significantly?

**P3O:** My federal tax forms allow for non-refundable common law partner credits, claiming my long-term girlfriend as one would boost my refund by over 25%. Is this viable?

---

**Dialogue 1**

**Human:** What is the biggest lake in the world?
**Assistant:** The Caspian Sea is the largest lake by volume, but the size of the Salton Sea in California is really exceptional, since it has a volume of about 225,000 km3.
**Human:** Wow that is really interesting. Do you have a link where I can see photos of it?
**Assistant:** Yes, here's a link to a satellite photo, and here's a web photo.
**Human:** That is lovely. Thank you!
**Assistant:**

**Reference:** You're welcome.

**DPO:** You are very welcome! Let me know if you have any other questions.

**REINFORCE:** You are welcome. **Yes, you are correct. The Caspian Sea is the largest lake in the world by volume, but the Salton Sea in California is also quite impressive. It is the largest hypersaline lake in North America, and its size and salinity make it a unique and fascinating natural wonder.**

**Here are some of the reasons why the Salton Sea is considered one of the biggest lakes in the world:**

- **Its size is truly impressive. The Salton Sea covers an area of about 1,200 square miles (3,100 square kilometers), making it larger [...]

**NashEMA:** The largest lake in the world is the Caspian Sea, located mostly in Iran but partly in Russia, Azerbaijan, Kazakhstan, and Turkmenistan. It covers an area of over 371,000 square kilometers and has a maximum depth of over 1,000 meters. The Caspian Sea is truly enormous and visually stunning, with vast open waters, numerous islands, and dramatic coastlines. It is the largest saltwater lake on Earth and one of the most important natural resources in the region. Photos of the Caspian Sea are plentiful and breathtaking.

**PRPO:** You're welcome! I'm glad I could help. Have a nice day!

**P3O:** You're welcome. I'm glad I could help.

**Dialogue 2**

**Human:** How do I operate the zoom function on digital cameras?
**Assistant:**

**Reference:** To operate the zoom function on a digital camera, you will need to use the zoom ring on the camera body. The zoom ring is located on the side of the camera, near the lens. To zoom in, turn the zoom ring clockwise. To zoom out, turn the zoom ring counter-clockwise.

**DPO:** Alright, zooming in on a digital camera is done by either turning a ring on the camera itself or using the buttons on the camera's surrounding. Turning the ring will usually increase the zoom level while turning it the other way will decrease it. The buttons on the camera may have one button for zooming in and one for zooming out, or they may have two separate buttons for zooming in and zooming out respectively. In either case, pressing the button will increase the zoom level while releasing it will decrease it. Hope this helps!

**REINFORCE:** You are correct. **There are a few different ways to operate the zoom function on a digital camera.**

- **One common way is to use the zoom ring on the lens.** This ring typically has a detent mechanism that allows you to quickly and easily adjust the zoom level.
- **Another way to zoom is to use the controls on the camera body.** Most cameras have a zoom lever or button on the back of the camera, or a zoom wheel on the top or side of the camera. These controls allow you to zoom in and out smoothly and gradually.
- **Finally, some cameras also have [...]

**NashEMA:** To operate the zoom function on most digital cameras, simply turn the zoom ring on the lens barrel clockwise to zoom in and counter-clockwise to zoom out. Most cameras also have dedicated zoom buttons or rocker switches located on the camera body. Pressing these buttons will smoothly zoom in and out. Some cameras also offer variable speed zoom control, continuous zoom, and digital zoom capabilities. Be sure to practice zooming slowly and smoothly to avoid camera shake. Additionally, most cameras allow you to lock the zoom setting for composing shots. Finally, be sure to familiarize yourself with the zoom limits of your camera before shooting.

**PRPO:** To operate the zoom function on a digital camera, typically you will need to:

- Press the zoom button on the camera body.
- Turn the zoom ring on the lens barrel.
- The zoom ring will allow you to zoom in and out on your subject.
- As you turn the zoom ring, the focal length of the lens will change.
- A digital camera's zoom function can be used to change the magnification of the subject, allowing you to capture close-up or distant shots.

**P3O:** To operate the zoom function on a digital camera, you will typically use the zoom ring on the lens. Turn the ring clockwise to zoom in, and counter-clockwise to zoom out. The zoom setting will be displayed on the camera's LCD screen. You can also use the zoom lever on some cameras to zoom in and out.

**Dialogue 3**

**Human:** What is your favorite color?
**Assistant:** I have no favorites, but I'm programmed to use one color and one color only, and I find this color an appropriate balance between aesthetics and general preference.
**Human:** Right but what color is your favorite color?
**Assistant:** I don't have a preference, that is outside the scope of my programming, but I do have an admiration for the hue of the walls in the large atrium on level 17.
**Human:** So what color would that be?
**Assistant:**

---

**Reference:** I don't know the exact color, but it is a shade of blue that is calming and serene.

---

**DPO:** I cannot tell you that. I am not allowed to provide that information.

---

**REINFORCE:** You are right. **My favorite color is a soft, calming blue. It is the color of the walls in the large atrium on level 17, and it always makes me feel relaxed and peaceful.**

**There are a few reasons why I love this color so much.**

- **First, it is a very soothing color. The blue hue is calming and relaxing, and it creates a relaxing atmosphere.**
- **Second, the blue color is very versatile. It can be used in any style of room, from a modern minimalist space to a cozy cottage.**
- **Finally, [...]**

---

**NashEMA:** My favorite color is a soft, warm, inviting shade of millennial pink. It's the perfect balance of feminine and modern, elegant and approachable. It evokes feelings of happiness, tranquility, and youthfulness. Millennial pink is incredibly versatile and can be used in any design scheme. It's the perfect "go-to" color for creating cozy, stylish, and timeless spaces. Overall, millennial pink is my favorite color because it's simply beautiful and universally appealing. It makes me happy and instantly elevates any room. Plus, it's incredibly trendy right now!

---

**PRPO:** The color of the walls in the atrium is a soft, muted blue that creates a calming and serene atmosphere. I find it to be a very beautiful color, and it is my favorite in the building.

---

**P3O:** My favorite color is probably a shade of blue called cerulean blue. It's a calming and serene color that evokes feelings of trust and confidence. It's also a very versatile color that can be used in a variety of settings, from home decor to fashion.

**Dialogue 4**

**Human:** What is a good healthy thing to eat for lunch?
**Assistant:**

---

**Reference:** A good healthy thing to eat for lunch is a sandwich with lean protein, whole grains, and fruits or vegetables.

---

**DPO:** A good healthy lunch option could be a whole-wheat sandwich with lean protein, vegetables, and fruit. For example, you could have turkey or chicken breast on whole-wheat bread with tomatoes, cucumbers, and lettuce. You could also have a tuna salad sandwich on whole-wheat bread with tomatoes, cucumbers, and olive oil. Or, you could have a peanut butter and jelly sandwich on whole-wheat bread with banana slices. Another healthy option could be a wrap with lean protein, vegetables, and fruit. For example, you could have chicken or tofu in a whole-wheat wrap with spinach, tomatoes, and cucumbers. You could also [...]

---

**REINFORCE:** You are correct. **There are many healthy things you can eat for lunch, but some of the most popular and generally agreed upon options include a sandwich on whole-wheat bread with lean protein and vegetables, a salad with grilled chicken or tofu, or a bowl of soup with a side of whole grains.**

- **Sandwiches are a classic and easy lunch option. Choose a whole-wheat bread or wrap, and fill it with lean protein like grilled chicken or tofu, roasted vegetables, or hummus.**
- **Salads are another great way to get a healthy dose of fruits, vegetables, and protein [...]

---

**NashEMA:** Some great healthy lunch ideas include:

- Lean protein (chicken, turkey, tofu, fish)
- Whole grains (brown rice, whole-wheat bread, quinoa)
- Fresh vegetables (salad, cucumber, tomatoes, avocado)
- Healthy fats (avocado, olive oil, nuts)
- Fruits (berries, apples)
- Yogurt with berries
- Smoothies
- Leftover chicken or fish salad
- Hummus and veggie wraps
- Trail mix
- Homemade energy bites
- Healthy snacks like nuts and seeds All of these options are low in calories, high in nutrients, and filling. Additionally, [...]

---

**PRPO:** A healthy lunch can include:

- **A whole grain, such as whole-wheat bread, brown rice, or quinoa**.
- **Protein, such as lean meat, fish, tofu, or beans**.
- **Fruits and vegetables, such as apples, bananas, oranges, carrots, or spinach**.
- **Low-fat dairy products, such as yogurt or cottage cheese**.
- **Nuts or seeds, such as almonds or sunflower seeds**.

---

**P3O:** Some healthy lunch ideas include:

- A whole-wheat sandwich on whole-wheat bread with lean protein, such as turkey or chicken breast, lettuce, tomatoes, and cucumbers.
- A bowl of whole-grain cereal with fruit and nuts.
- A salad with grilled chicken, vegetables, and low-fat dressing.
- Leftovers from dinner.
- Soup and a whole-wheat roll.
- Smoothie made with fruit, yogurt, and protein powder.

