# OpenReview forum: "Mitigating Preference Hacking in Policy Optimization with Pessimism"
_TMLR — Rejected by TMLR_

### Review · Reviewer_b3DE · 2026-02-07

**Summary Of Contributions:**

*Summary*

The paper proposes a pessimism-based objective for preference optimization that explicitly accounts for uncertainty in the learned preference function, and argues that prior pessimistic approaches can behave pathologically when the opponent is unrestricted under coverage gaps. Two algos, P3O as a practical min–max procedure and PRPO as BLT variant, are introduced. Empirical evidence over TL;DR task and helpfulness task is presented.

---

*Strength*

1. Method is theoretical-grounded and with guarantee. The theoretical claim strengthened the credibility of the method.
2. Motivation is clear, with adequate discussion over related pessimistic online&offline works. Also, the contribution of the work is well positioned to my knowledge. **However, I am not very familiar with some of the pessimistic baselines discussed in this work. Therefore, I cannot guarantee whether the contributions proposed in this work is fully original.**
3. There are empirical evidence.

---

*Weakness*
1. The empirical evaluation is somehow weak. See my comment in later section for details.
2. There are many hyperparameters introduced, but I didn't find systematic ablation.

**Audience:**

Yes

**Audience Explanation:**

Yes,

The formulation is interesting, and many audience working over online RL will be benefited from reading this.

**Broader Impact Concerns:**

Not applied here.

**Claims And Evidence:**

Yes

**Claims Explanation:**

Partially, yes.

The theory supports a restricted robustness claim cleanly, and the experiments support reduced hacking under their evaluator+symptom metrics, but the evaluation choice and the proxying of coverage in the algorithm leave room for alternative explanations. More empirical evidence and systematic evaluations should be presented to make the evidence clearer.

**Requested Changes:**

- Please check all the citations again; don't misuse \citep and \citet.
- I'm curious about offline baselines (e.g., Huang et al., 2025). How does your method compare to, or outperform, simply adding a $\chi^2$ regularization term?
- I think the empirical evaluation setup should be strengthened. For example, on the summarization benchmark, the current evaluation relies only on win rate (WR) judged by Gemini 1.5, which is not a standard protocol for this benchmark. I skimmed some recent works on this task (e.g., arxiv.org/pdf/2602.02495), where evaluation is done using a trained judge/scoring model (see Appendix C.2 for details). Also see Appendix E of arxiv.org/pdf/2406.18853. I suggest following the evaluation setup from these works. LLM-as-a-judge can be fragile, especially when using a relatively weak Gemini variant. If budget permits, please also report WR using a stronger judge model.
- I'm confused about why you use T5-XL 3B as the base model. There are more standard 3B baselines today—e.g., Llama-3.2-3B-Instruct or Qwen2.5-3B. Why not run your method on these models? If computation budget allows, I strongly encourage reporting results on these more widely used models.
- **Since TMLR allows appendices, I suggest moving the appendix material from the supp into the main paper to improve readability.**
- Appendix H is helpful. Could you add a short paragraph explaining *why* the preferred responses are better in those examples (i.e., what specific qualities the method improves)?
-  If I understand correctly, from def 3.1, $\epsilon$ is truly an L1 bound, the natural scaling you can derive from def 3.1 is closer to $\sqrt{C}\cdot\sqrt{\mathbb E_{\pi_{\text{data}}}[(p_1-p_2)^2]}$, and then you still need to relate $\mathbb E[(p_1-p_2)^2]$ to $\mathbb E|p_1-p_2|$. Since $|p_1-p_2|\le 1$, you get $\mathbb E[(p_1-p_2)^2]\le \mathbb E|p_1-p_2|$, which tends to produce something like $\sqrt{C\epsilon}$ instead of $\sqrt{C}\epsilon$ in Lemma 3.2. Please clarify this point.

---

> ### Author Response · Authors · 2026-03-06
> **Reply to Reviewer b3DE (Part 1/2)**
>
> We would like to thank the reviewer for their time and thoughtful comments. We appreciate that the reviewer found the method theoretically grounded, the motivation clear, and the contribution well-positioned. We address the specific concerns below.
>
> **On the empirical evaluation setup and LLM-as-a-judge:**
>
> Please also see our response to Reviewer GTXY regarding the scale of Gemini 1.5 Flash, the prohibitive cost of using Gemini 2.5 Pro, and our two-fold evaluation approach.
>
> We appreciate the references to recent works (e.g., arXiv:2602.02495, arXiv:2406.18853) that use trained scoring models for evaluation. Reviewing the appendices of those specific papers reveals that they rely on much smaller models as judges, such as the GPT-2 based `Tristan/gpt2_reward_summarization`. The more concerning aspect about the above papers is the use of a much stronger model as a policy model (i.e., Llama, Qwen) being evaluated against smaller reward models, which are well known to be easily hacked (especially in cases where the policy model is bigger than the reward model).
>
> While the use of trained judges for evaluation is a common paradigm, it introduces a critical vulnerability: the trained reward models—especially smaller ones like GPT-2—are typically susceptible to the same spurious correlations, OOD errors, and biases present in the preference datasets they were trained on (in this case, TL;DR). Using them as a "gold-standard" evaluator to measure over-optimization of preferences risks falling into a trap where policies demonstrating hacking behavior might score higher. Because the policy exploits the same dataset biases that the trained judges also learned, it becomes exceedingly difficult to isolate the actual effect of hacking.
>
> Instead of relying on a potentially biased, in-domain trained reward model, our evaluation leverages a frontier LLM (Gemini 1.5 Flash) as a potentially more capable, out-of-distribution, third-party proxy (Zheng et al., 2023). Most importantly, we do not rely on this proxy alone. By combining this LLM with evaluator-independent metrics (Figure 3: response length, list fraction), we demonstrate that our method can actively mitigate the real-world hacking behaviors present in the original datasets.
>
> **On the choice of base models (T5-XL 3B vs Llama-3.2/Qwen2.5):**
>
> The choice of T5 and PaLM-based architectures for the policy and reward models was to directly replicate the experimental setup of established studies on reward hacking for these specific datasets (e.g., Eisenstein et al., 2024; Fisch et al., 2024). This allows us to cleanly isolate the effect of our algorithmic contribution (pessimism) from architectural confounders, ensuring a fair comparison of objective functions without introducing the innate biases of newer instruction-tuned models. We will add a discussion clarifying this design choice.
>
> **On comparisons to offline baselines (e.g., $\chi^2$ regularization):**
>
> Thanks for pointing out this baseline. Adding a $\chi^2$ penalty (like in Huang et al.) is definitely a strong way to regularize the policy offline. However, $\chi^2$ offers a static way to regularize the policy in the offline setting, and provides no way to account for the actual uncertainty in the reward or preference model that arises from limited preference data. P3O, on the other hand, is designed to be explicitly uncertainty-aware. Instead of applying a blind penalty everywhere, the max-min objective aims to identify regions of high uncertainty and penalize the policy in those regions. Additionally, the above penalties are typically designed for standard reward models (i.e., the Bradley-Terry-Luce setting), whereas P3O is designed to handle general pairwise preferences. We will add a discussion to the related works section clarifying this distinction.
>
> **On formatting, citations, and missing appendices:**
>
> We apologize for the confusion regarding the appendices! According to TMLR submission guidelines, we included the appendices (containing hyperparameter ablations, qualitative examples, and proofs) in the supplementary `.zip` file, making them easy to miss. We will merge the appendices into the main PDF to improve readability. We will also carefully audit our use of `\citep` vs. `\citet`.
>
> **Examples in Appendix H:**
>
> We are glad that the examples in Appendix H were helpful. We would like to point out the introductory paragraph of Appendix H, which explains why the responses from P3O/PRPO are better. It details how baselines (like DPO, REINFORCE, and NashEMA) tend to generate overly long, extractive, or repetitive list-heavy responses, while our methods maintain concise and naturally formatted outputs.

---

> > ### Author Response · Authors · 2026-03-06
> > **Reply to Reviewer b3DE (Part 2/2)**
> >
> > **Lemma 3.2:**
> >
> > Thank you for checking the math. The intuition is correct, and our formulation aligns with it. Lemma 3.2 explicitly states the bound as $2\sqrt{C\epsilon}$, rather than $2\sqrt{C}\epsilon$. As detailed in the proof in Appendix B.1, we arrive at this by bounding the squared difference using the Hellinger distance, $H^2(p, p^*)$. This step naturally relates the L2 distance to the total variation (the L1 bound, $\epsilon$), which yields the $\sqrt{C\epsilon}$ scaling in the final bound.
> >
> > **Hyperparameter Ablation:**
> >
> > Appendix G details the different hyperparameters and the values that were swept over. For the standard RLHF baselines, the primary hyperparameter that required sweeping was $\beta$ for the KL penalty to prevent reward hacking (Appendix G.4, Figure 8). In contrast, we fixed $\beta$ for P3O to a standard small value ($10^{-5}$) and primarily tuned the regularization for the preference model likelihood (i.e., $\lambda$) and the learning rate for the preference model (i.e., $\eta_p$). We found these values to generalize to both tasks—we found these parameters on TL;DR and didn't change them when performing experiments on Helpfulness. We will add a discussion on the implications of these parameters in the Appendix, but primarily, the $\lambda$ parameter tends to be on the larger side to simulate an uncertainty set of preference functions closely tied to the preference dataset.
> >
> > **References**
> >
> > [1] Eisenstein et al. *Helping or Herding? Reward Model Ensembles Mitigate but do not Eliminate Reward Hacking.* COLM, 2024.
> >
> > [2] Fisch et al. *Robust Preference Optimization through Reward Model Distillation.* arXiv:2405.19316, 2024.
> >
> > [3] Huang et al. *Correcting the Mythos of KL-Regularization: Direct Alignment without Overoptimization via Chi-Squared Preference Optimization.* arXiv:2407.13399, 2024.
> >
> > [4] Zheng et al. *Judging LLM-as-a-Judge with MT-Bench and Chatbot Arena.* NeurIPS, 2023.

---

> > > ### Comment · Reviewer_b3DE · 2026-03-09
> > >
> > > Thanks for the response. I think the following concerns remain unresolved, though these reflect my own view, and I will defer the final judgment to the AE.
> > >
> > > On the use of Gemini-1.5 Flash as the judge for all WR evaluations: I remain concerned about this choice. Gemini-1.5 Flash no longer appears in the current API documentation (I couldn't even find it as https://ai.google.dev/gemini-api/docs/pricing), which suggests that it may already be an outdated model for use. There are many stronger and more competitive judge models available at similar token cost, including models accessible through OpenRouter (e.g. GPT-series or Qwen-series models). Given the well-known fragility of LLM-as-a-judge evaluations, relying on an older and relatively weak judge substantially weakens the empirical evidence. After reading the other reviews, I think that this is not just mine concern, which is raised by the other reviewers as well.
> > >
> > > On the model choice: I agree that replicating prior experimental setups is useful for comparability. However, this does not fully address my concern, which is whether the conclusions remain valid under stronger and more modern models.
> > >
> > > I do not have additional concerns beyond these. As noted above, these are simply the points that remain from my perspective. If the other reviewers and the AE do not consider them major issues, I am totally fine with it.

---

> > ### Author Response · Authors · 2026-03-09
> > **Thanks for the comment**
> >
> > We acknowledge your concerns, but want to reemphasize also that the quantitative metrics tell a fairly clear story (in terms of length or list fraction of the responses), in addition to what can be considered qualitatively subjective LLM-as-a-judge evaluation. We also invite the reviewer to check some of the examples included in the supplement. The stylistic differences between the different approaches are quite apparent. We are, unfortunately, unable to run more experiments with other base models or other evaluators at this point, but we do believe that taken collectively, these all point to a fairly consistent conclusion.

---

### Review · Reviewer_PVgr · 2026-02-10

**Summary Of Contributions:**

**Disclaimer: I am not a specialist of RLHF (but rather of sample complexity in RL). I did my best to read the submission, which I should perhaps have asked not to review.**

The submission studies preference-based policy optimization, i.e., how, given a data set of preference data (contexts and associated pairs of observations, with the indication of which is the best) and a reference policy, one can construct a new policy aligning well with the historical preferences and not too far from the reference policy. This is achieved by optimizing some quantitative criteria, typically of the form max-min of some measure of preference $J(\pi,\pi')$, the argument of the maximum over $\pi$ being the policy built. The article revolves around what to take as $J(\pi,\pi')$. Previous approaches computed $J$ based on an MLE estimate of the preference function, and this led to overfitting issues. More recent approaches use as $J(\pi,\pi')$ some minimum of quantities of the $J_P(\pi,\pi',p)$, where the preference functions are in some restricted class around the ground truth $p^\star$. The analysis of this approach relies on a condition called unilateral coverage, which is claimed to be strong and to lead to issues (at least on a specific example), which motivates the introduction of a new concept of covered policy set to also restrict the class of comparators $\pi'$.

Performance guarantees are exhibited (Lemma 3.2) on the outcome of the corresponding optimization, but the procedure implemented in simulations follows from various relaxations and twists on this procedure, to make it computationally more tractable.

**Additional Comments:**

I list my remarks in the order of the document:
- Page 3, can there be ties between $y_1$ and $y_1$? The setting does not seem to allow for that.
- Equation (3): some reference / details for the existence of the Nash equilibrium would be great (does Sion's lemma apply here, as I believe?)
- Page 4: the notation $p^\star$ is not defined yet (that this is the ground truth is only indicated on page 6; it would be useful to formally define what the ground is, and BTW, also, on page 3, how available preference data was sampled, etc.)
- Page 4: The set $\mathcal{P}$ depends on $p^\star$, which is unknown I imagine, so i guess that (6) is like an idealized procedure that cannot be implemented (and not only because of computational issues but rather because it depends on unknown quantities)
- Page 4: I think it would be great to formally restate the definition of unilateral coverage
- Page 5: The middle paragraph 'This example highlights... a remedy for this' conveys nice intuitions and insights
- Page 5: I did not get what the sampling policy $\pi_{\mbox{data}}$ was, nor whether it was known to the learner
- Page 5, Equation (8): same issue here, does this policy depend on quantities unknown to the learner?
- Lemma 3.2: I'm unsure about its meaning---and I guess I would recommend explain better on page 3 what the objectives are: e.g., output a policy $\hat{\pi}$ such that $J(\hat{\pi},\pi')$ is larger or close to $1/2$ for all $\pi'$? Again, I struggle to follow the content because I'm probably not competent enough
- Sentence after Lemma 3.2: Would it be really more complicated to keep contexts in the main body, is it really useful to defer the treatment of contexts to Appendix B.1? I guess I will know when I see it
- Footnote 3: Could be put after some formal re-statement of the definition of unilateral coverage
- Equations (8) vs. (9): I could be useful to explain better how (9) was obtained from (8); I could follow the logic of most of the other relaxations / twists on pages 7-8
- Experiments: they are even more difficult for me to evaluate (especially without the appendices)

**Audience:**

Yes

**Audience Explanation:**

The topic is clearly timely and important, with lots of work going on RLHF and the mitigation of preference hacking

**Broader Impact Concerns:**

I might have expected a Broader Impact Statement section but did not see any

**Claims And Evidence:**

No

**Claims Explanation:**

The main body seems generally clear enough (see below for suggestions to make it more accessible to laymen like me)... but I believe that the the authors forgot to attach the appendices to their submission! This is an easy-to-correct issue. The text of the main body is clear enough about what should be contained in these appendices, and I can well imagine their content.

**Requested Changes:**

Again, sorry for rather providing an informed guess than a deep review. I will read closely what other reviewers write and do my best.

---

> ### Author Response · Authors · 2026-03-06
> **Reply to Reviewer PVgr**
>
> We would like to thank the reviewer for their careful reading and thoughtful comments. We are glad the reviewer found the intuitions behind the method clear and the topic timely.
>
> **Missing appendices:**
>
> Firstly, we apologize for this confusion! Following TMLR's submission guidelines for supplementary material, our appendices (which contain the proofs for Lemma 3.2, derivations, and tabular experiments) were uploaded in the supplementary `.zip` file rather than the main PDF, making them easy to miss. We completely agree that the paper is much easier to read as a single document. We will merge the appendices directly into the main PDF.
>
> **On mathematical and notational clarifications:**
>
> Thank you for highlighting these points; addressing them will definitely help us improve the clarity of our theoretical explanation. We will make the following updates to the text:
>
> **Ties (Page 3):** Standard RLHF preference datasets generally assume strict preferences, i.e., ($y_w > y_l$). However, ties are already accommodated in our formulation, where ties would be represented as $p(y_1, y_2) = 0.5$. We will add a clarifying footnote.
>
> **Nash Equilibrium Existence (Eq 3):** You are correct that Sion's minimax theorem applies here due to the compact and convex policy spaces and the strictly concave-convex structure of the objective introduced by the KL terms. However, we want to clarify that this specific Nash objective in Eq. 3 is not a novel contribution of our paper, but is adopted directly from prior works. We will update the main text to explicitly reference Munos et al. (2024) and Swamy et al. (2024) for the formal guarantees regarding the existence of the Nash equilibrium in this setting.
>
> **Defining $p^*$ and $\pi_{\text{data}}$:** We will formally define the ground-truth human preference $p^*$ earlier in Section 3. We will also clarify that $\pi_{\text{data}}$ is the unknown behavior policy that generated the offline dataset (which in practice would be approximated by performing SFT on the preference dataset).
>
> **Idealized procedure in Eq 6 & 8:** While our theoretical guarantees (Lemma 3.2) require that $\mathcal{P}$ contains the unknown $p^*$, a set satisfying this condition can be constructed entirely in a data-dependent manner. We do not need knowledge of any ground-truth quantities; instead, we can use standard learning-theoretic guarantees to build a confidence region around the empirical estimate. We will clarify this in the revision and provide a concrete instantiation of such a definition.
>
> **Meaning of Lemma 3.2 & Unilateral Coverage:** We will introduce the definition of unilateral coverage into the main text. Intuitively, Lemma 3.2 is the preference-based analogue to guarantees in offline RL: essentially, the lemma guarantees that the policy obtained from our pessimistic objective will not be "dispreferred" to any other policy whose actions stay within the offline data distribution, up to an error bound $2\sqrt{C \epsilon}$.
>
> **Equations 8 vs 9:** Equation 9 introduces a relaxation of the hard membership constraint ($\pi' \in \Pi(\pi_{\text{data}}, C)$) from Equation 8 by introducing a regularization term. Instead of strictly bounding the min-player's policy class (which in practice is quite hard for large models), we apply a soft penalty via the KL divergence. We will add a transition sentence to make this logical derivation explicitly clear.
>
> **Contexts in the main body:** While we appreciate the suggestion, we decided to defer the treatment of contexts to the appendix. Including the context variable throughout the main body may make the main text equation-heavy without fundamentally altering the core logic or changing the structure of the proofs.
>
> **On Broader Impact:**
>
> We will add a Broader Impact Statement at the end of the manuscript, discussing how mitigating reward hacking can lead to safer and more reliable AI alignment.
>
> **References**
>
> [1] Munos et al. *Nash Learning from Human Feedback.* ICML, 2024.
>
> [2] Sion. *On General Minimax Theorems.* Pacific Journal of Mathematics, 1958.
>
> [3] Swamy et al. *A Minimaximalist Approach to Reinforcement Learning from Human Feedback.* ICML, 2024.

---

> > ### Comment · Reviewer_PVgr · 2026-03-06
> > **Note that an updated PDF can be submitted**
> >
> > I thank the authors for their responses. It turns out that reading the other reviews, I had located and read meanwhile the appendices.
> >
> > Some answers, like $\mathcal{P}$ containing the unknown $p^*$ through a specific construction, are of course difficult to grasp without the details.
> >
> > But if am not mistaken, the TMLR policy allows, and even encourages, submitting a revised PDF at this stage of the reviewing process.

---

> > > ### Author Response · Authors · 2026-03-08
> > > **Updated PDF submitted**
> > >
> > > Thank you for the suggestion. We have uploaded an updated PDF with all the proposed changes in red to OpenReview.

---

### Review · Reviewer_GTXY · 2026-02-20

**Summary Of Contributions:**

### Summary

This paper addresses the problem of reward/preference hacking in RLHF, where a policy exploits vulnerabilities in a reward/preference model trained on limited data. The authors identify a key issue with standard pessimistic approaches: an unconstrained opponent can force the policy to generate uncertain, out-of-distribution actions. To solve this, they propose a novel "restricted pessimistic Nash" objective, where the opponent is constrained to a set of policies covered by the preference data. Then, the authors derive two practical algorithms, P3O (preference-based) and PRPO (reward-based), to implement this idea through an adversarial max-min objective. Experiments on summarization and helpfulness tasks show that these methods effectively mitigate preference hacking and achieve higher win rates compared to standard baselines like DPO and REINFORCE


### Strengths

1.	The paper is well motivated. It recognizes the "unconstrained opponent" problem and explains why simple pessimism fails. The proposed "restricted pessimistic Nash" objective is principled and theoretically sound for this specific problem.

2.	The authors translate a complex theoretical objective into practical algorithms (P3O/PRPO). The max-min game formulation is somehow intuitive: the policy player tries to win, while the preference player tries to find the policy’s weaknesses.

3.	The paper is well formatted with initial toy problems, theoretical analysis, followed by approximations and algorithms.


### Weaknesses
1.	The practical algorithms rely on several approximations to the original theoretical objective. For example, a hard constraint on the opponent's policy space is replaced by a soft KL penalty, and a variational upper bound is used to simplify the objective. The impact of these approximations is not fully analyzed.

2.	The max-min game structure is common in adversarial training (e.g., GAN), which is known to be susceptible to "mode collapse." The policy may learn to generate only a few safe types of responses, leading to a lack of diversity. The paper does not discuss or evaluate this potential risk.

3.	The evaluation is performed on two tasks: summarization and helpfulness. It is unclear how well the method would perform on other preference tasks, such as safety alignment.

4.	About LLM-as-a-judge: Gemini 1.5 Flash is a relatively old and small-size model, and it is unclear how well it can perform the judgment. The authors are encouraged to report the correlation between LLM judge and human expert (or ground truth), e.g., Kohen’s Kappa. If possible, use stronger models such as Gemini 2.5 Pro.

**Audience:**

Yes

**Audience Explanation:**

The paper focuses on preference learning and mitigating reward hacking, which are both important problems.

**Claims And Evidence:**

No

**Claims Explanation:**

The evaluation could be made more thorough and sound. Please see weakness 3 and 4 for details.

**Requested Changes:**

See weaknesses for details.

1.	Investigate mode collapse: It would strengthen the paper to include an analysis of the diversity of the generated responses. This is to show whether the policy's output diversity is maintained during the max-min training.

2.	If possible, please test the method on a task from a different domain (e.g., safety alignment) to demonstrate broader applicability. It is also good to try other benchmarks, such as Alpaca Eval 2.0.

3.	About the LLM judge: please consider reporting Kohen’s Kappa or reporting results using a more powerful model.

---

> ### Author Response · Authors · 2026-03-06
> **Reply to Reviewer GTXY**
>
> We would like to thank the reviewer for their time and constructive feedback. We appreciate the strengths the reviewer has identified, particularly recognizing the importance of the problem and finding our max-min formulation intuitive and theoretically sound. We address the specific concerns below.
>
> **On the impact of approximations (soft KL and variational bounds):**
>
> Translating the proposed restricted Nash objective requires approximations to make it tractable algorithmically. To evaluate the impact of these approximations on the quality of solutions, we conduct experiments on a tabular setting in Appendix F. In this setting, we evaluate the exact Nash objective (EP3O) and the approximate variant (P3O) using brute-force search. As shown in Figures 4 and 5 (Appendix F), we notice that the approximate objective closely tracks the exact objective (EP3O), and more importantly, preserves the core property, i.e., mitigating over-optimization and remaining in support of the sampling policy. We will add a paragraph discussing these experiments in Section 4 to highlight this finding.
>
> **On the risk of mode collapse and lack of diversity:**
>
> Thank you for raising this point; it is an important clarification regarding adversarial training objectives. In our formulation, the max-player is regularized towards the reference policy ($\pi_{\text{ref}}$) via a KL term, similar to standard RLHF. This regularization discourages entropy collapse and makes P3O no more susceptible to mode collapse than any standard RLHF method. In fact, not overfitting the reward function can actually be better for preserving response diversity. Standard RLHF methods often collapse into narrow, repetitive stylistic hacking behaviors (e.g., length hacking or list formatting). We see evidence that P3O avoids this in our empirical evaluation, where it is consistently preferred by a third-party judge. Furthermore, the preservation of natural response diversity is self-evident in the qualitative examples shown in Appendix H.
>
> **On the evaluation tasks:**
>
> This is a good point, and yes, the reason to focus on the two standard tasks of helpfulness and summarization was to focus on well-studied (and quite stubborn to prevent!) forms of preference hacking. The proposed algorithms are in no way specific to those tasks, however, and can be applied to other tasks with minimal changes (in fact, we use similar hyperparameters for both settings, with the only difference being model sizes and architectures). The only consideration that we rely on is gradient-based updates to our preference/reward model — extending our approach to other types of problem settings (e.g., safety alignment) is a great direction for future research.
>
> **On the LLM-as-a-judge evaluation:**
>
> To address the reviewer's concern around evaluation and the choice of LLM judge: in short, (1) using a frontier LLM-as-a-judge is standard methodology in recent literature (Eisenstein et al., 2024; Wu et al., 2024; Gao et al., 2024; Zheng et al., 2023), as they have been shown to correlate well with human preferences on average, (2) given our base policy and preference models are T5-Large (770M) and T5-XL (3B), Gemini 1.5 Flash is substantially bigger than our models and serves as a competent “expert”, while transitioning to Gemini 2.5 Pro will tend to be substantially more expensive (8x for input and 16x for output), and (3) we strongly believe that the combination of this subjective LLM-as-a-judge metric with our objective distributional metrics (e.g., mean length, frequency of lists) gives compelling and comprehensive proof of the efficacy of our method.
>
> More concretely, because conducting a human annotation study to report correlation metrics (like Cohen’s Kappa) on our specific generations is prohibitively expensive, we rely on the established precedent that frontier LLMs exhibit strong alignment with human judgments on these benchmarks (Zheng et al., 2023). That being said, we agree that looking only at LLM-as-a-judge metrics is insufficient for a conclusive evaluation, which is why we also evaluate the method on objective distributional metrics shown in Figure 3. These metrics—such as response length and list-style formatting—do not rely on any preference evaluator, allowing us to verify that a policy is not obviously “gaming” the reward signal in some idiosyncratic way.
>
> Importantly, the proposed objectives (P3O/PRPO) are in no way designed to explicitly target length or formatting, and they do not use this information during training. They are generic robust objectives intended to mitigate any form of reward/preference hacking that arises from limited preference data. The fact that P3O achieves high win-rates while simultaneously avoiding objective hacking symptoms is a strong signal that the learned policy is genuinely better. We will revise Section 5 to make this two-pronged evaluation strategy and our rationale for judge selection explicit in the paper.

---

> > ### Author Response · Authors · 2026-03-06
> > **References**
> >
> > [1] Eisenstein et al. *Helping or Herding? Reward Model Ensembles Mitigate but do not Eliminate Reward Hacking.* COLM, 2024.
> >
> > [2] Gao et al. *REBEL: Reinforcement Learning via Regressing Relative Rewards.* NeurIPS, 2024.
> >
> > [3] Wu et al. *Self-Play Preference Optimization for Language Model Alignment.* arXiv:2405.00675, 2024.
> >
> > [4] Zheng et al. *Judging LLM-as-a-Judge with MT-Bench and Chatbot Arena.* NeurIPS, 2023.

---

> > ### Comment · Reviewer_GTXY · 2026-03-18
> >
> > I would like to thank the authors for their response. My concern regarding mode collapse is addressed. However, I hold my points for evaluation, including insufficient evaluation tasks and the weak judge. As mentioned by Reviewer b3DE, maybe you could follow arxiv.org/pdf/2602.02495 to use tuned open-source judge models, at least for the summarization task.

---

### Author Response · Authors · 2026-03-08
**Summary of Revisions to PDF**

We thank the reviewers for their constructive feedback. All changes in the revised manuscript are highlighted in **red** for easy identification. Below is a summary of the key revisions:

**Added the Appendix.** We have attached the appendix to the main paper.

**Notation and clarity (Section 2–3).** We added early definitions of $p^\star$ and $\pi_{\mathrm{data}}$, a footnote on ties, a reference for Nash equilibrium existence, and clarified that the uncertainty set $\mathcal{P}$ can be constructed from data (citing Zhu et al., 2023; Zhan et al., 2024). We added a formal definition of unilateral coverage in Appendix B.1 and an intuitive explanation of Lemma 3.2's connection to single-policy concentrability guarantees in offline RL (Xie et al., 2021; Rashidinejad et al., 2022). We also added a transition paragraph motivating the relaxation from the hard constraint to the soft KL penalty.

**Approximation quality (Section 4).** We added a paragraph discussing how the tabular experiments in Appendix F validate that the approximate P3O objective closely tracks the exact one.

**Evaluation and experiments (Section 5 and Appendix).** We expanded the evaluation protocol to explain the two-pronged strategy (LLM judge + evaluator-independent metrics). Discussions of model choice justification and output diversity / mode collapse were added to a new Appendix section.

**Literature review (Section 6).** We added a discussion to $\chi^2$-regularization (Huang et al., 2024) in the pessimism paragraph.

**Broader Impact (Section 7).** Added a Broader Impact statement discussing how evaluators trained on limited data may score harmful outputs favorably, and how pessimistic objectives help mitigate this.

**Citations.** Added references: Rashidinejad et al. (2022), Zhan et al. (2024), Zheng et al. (2023).

---

> ### Comment · Reviewer_PVgr · 2026-03-09
> **Thank you --- well received!**
>
> Thanks for preparing this revised PDF, we received it well.

---

### Decision · Action_Editor_PYN7 · 2026-03-26

**Recommendation:** Reject

**Additional Comments:**

The topic of this paper is more robust RLHF. The key idea is to prevent overfitting to reward or preference models using pessimism. The authors develop theory, propose practical algorithms, and evaluate them empirically. The evaluation is done on TL;DR summarization and Anthropic helpfulness datasets.

This paper was well received by the reviewers. The authors addressed all comments but two: outdated models (T5 XL 3B) and outdated LLM judges (Gemini 1.5 Flash). I agree with the authors that older models allow for a fair comparison to prior works. However, without more modern models, the relevance of this work to modern generative AI is in question. I suggest that the authors incorporate more modern models and judges, and resubmit. Other than that, this is a great paper for TMLR.

**Audience:**

Yes

**Audience Explanation:**

RLHF and post-training are popular topics. It is also well known than RL with generative AI models is susceptible to reward hacking. Therefore, this paper is timely and would have audience.

**Claims And Evidence:**

No

**Claims Explanation:**

This paper proposes new RLHF algorithms and evaluates them empirically. The theory and derivations of the algorithms are solid. The issue is empirical evaluation. The paper uses outdated models (T5 XL 3B) and outdated LLM judge (Gemini 1.5 Flash). I agree with the authors that older models allow for a fair comparison to prior works. However, without more modern models, the relevance of this work to modern generative AI is in question. Both Llama and Qwen have modern 3B models, and many LLM judges (both open and closed) exist. This was the main concern of two reviewers and the main reason for leaning towards a rejection.

**Resubmission Of Major Revision:**

The authors may consider submitting a major revision at a later time.